# SCNode: Spatial and Contextual Coordinates for Graph Representation Learning

**Md Joshem Uddin**                                                    *mdjoshem.uddin@utdallas.edu*
*Department of Mathematical Science*
*The University of Texas at Dallas*

**Astrit Tola**                                                                          *atola@fsu.edu*
*Department of Mathematics*
*Florida State University*

**Varin Singh Sikand**                                                      *varin.sikand@utdallas.edu*
*Department of Computer Science*
*The University of Texas at Dallas*

**Cuneyt Gurcan Akcora**                                                      *cuneyt.akcora@ucf.edu*
*AI Initiative*
*University of Central Florida*

**Baris Coskunuzer**                                                          *coskunuz@utdallas.edu*
*Department of Mathematical Science*
*The University of Texas at Dallas*

**Reviewed on OpenReview:** *https://openreview.net/forum?id=wdcdKeFbfQ*

## Abstract

Effective node representation lies at the heart of Graph Neural Networks (GNNs), as it directly impacts their ability to perform downstream tasks such as node classification and link prediction. Most existing GNNs, particularly message passing graph neural networks, rely on neighborhood aggregation to iteratively compute node embeddings. While powerful, this paradigm suffers from well-known limitations of oversquashing, oversmoothing, and underreaching that degrade representation quality. More critically, MPGNNs often assume homophily, where connected nodes share similar features or labels, leading to poor generalization in heterophilic graphs where this assumption breaks down.

To address these challenges, we propose *SCNode*, a *Spatial-Contextual Node Embedding* framework designed to perform consistently well in both homophilic and heterophilic settings. SCNode integrates spatial and contextual information, yielding node embeddings that are not only more discriminative but also structurally aware. Our approach introduces new homophily matrices for understanding class interactions and tendencies. Extensive experiments on benchmark datasets show that SCNode achieves superior performance over conventional GNN models, demonstrating its robustness and adaptability in diverse graph structures.

## 1 Introduction

Over the past decade, GNNs have made a breakthrough in graph machine learning by adapting several deep learning models originally developed for domains such as computer vision and natural language processing to the graph setting. By using different approaches and architectures, GNNs produce powerful node embeddings by integrating neighborhood information with domain features. While recent GNN models consistently

outperform the state-of-the-art (SOTA) results, in the seminal papers (Xu et al., 2019; Morris et al., 2019), the authors showed that the expressive power of message passing GNNs is about the same as a well-known former method in graph representation learning, i.e., the Weisfeiler-Lehman algorithm (Shervashidze et al., 2011). Furthermore, recent studies show that GNNs suffer from significant challenges such as oversquashing (Topping et al., 2021), oversmoothing (Oono & Suzuki, 2020), and underreaching (Barceló et al., 2020).

Another major issue with MPGNNs is their reliance on the (useful) homophily assumption, where edges typically connect nodes sharing similar labels and node attributes (Sen et al., 2008). However, many real-world scenarios exhibit heterophilic behavior, such as in protein and web networks (Pei et al., 2019), where conventional GNNs (Hamilton et al., 2017; Veličković et al., 2018a) may experience significant performance degradation, sometimes even underperforming compared to a multilayer perceptron (MLP) (Zhu et al., 2020; Luan et al., 2022; 2024). Therefore, there is a pressing need to develop models that function effectively in both homophilic and heterophilic settings.

We attribute current GNN shortcomings to their inability to capture two complementary sources of information in graphs: *spatial information*, which reflects the class distribution in a node's immediate neighborhood, and *contextual information*, which measures a node's relation to class representatives across the entire graph. To address this, we position each node relative to all classes. In the attribute space, we introduce *class landmarks*, reference embeddings that act as beacons for each class, and record the distances from each node's feature vector to these landmarks as its contextual coordinates. In the graph topology, we encode spatial coordinates by computing the distribution of neighboring node labels within each node's local subgraph.

Our landmark scheme establishes an implicit coordinate system: by triangulating distances to class landmarks, we locate each node in attribute space much like a GPS. These relative coordinates capture global patterns that standard message-passing GNNs often miss due to underreaching, i.e., the inability to propagate sufficient information from distant but relevant nodes within a limited number of layers. Spatial embeddings encode local label distributions in multi-hop neighborhoods, enriching high-frequency structural signals, while contextual embeddings inject low-frequency global components through landmark-based coordinates. From a spectral perspective, concatenating these embeddings expands the effective receptive field by combining complementary frequency bands. This joint representation mitigates the 1-WL locality bottleneck and enhances expressivity by unifying local structural distributions with long-range semantic context.

By leveraging relative positioning, we generate feature vectors that seamlessly blend spatial and contextual cues. The resulting *SCNode* model achieves state-of-the-art performance on node classification and link prediction across both homophilic and heterophilic benchmarks. Furthermore, SCNode embeddings can be dropped into any GNN as plug-and-play components, remedying underreaching and consistently boosting performance in diverse graph settings.

**Our contributions** can be summarized as follows:

- We propose a novel approach to capture *the relative positioning of nodes* with respect to node classes, incorporating both spatial and contextual perspectives.

- We introduce *class-aware homophily matrices*, providing detailed insights into homophily tendencies and enabling a deeper understanding of class interactions in the graph.

- Our SCNode model effectively integrates spatial and contextual information, *delivering state-of-the-art performance* in both node classification and link prediction tasks.

- We demonstrate that SCNode vectors can be easily integrated into any GNN model as *plug-and-play components*, *significantly boosting performance* in both homophilic and heterophilic settings, showcasing their versatility across diverse real-world applications.

## 2 Related Work

Over the past decade, GNNs have dominated graph representation learning, especially excelling in node classification tasks (Xiao et al., 2022). After the success of Graph Convolutional Networks (GCN) (Kipf

& Welling, 2017), using a neighborhood aggregation strategy to perform convolution operations on graph, subsequent efforts focused on modifying, approximating, and extending the GCN approach, e.g., involving attention mechanisms (GAT) (Veličković et al., 2018a), sampling and aggregating information from a node's local neighborhood (GraphSAGE) (Hamilton et al., 2017). However, a notable scalability challenge arose due to its dependence on knowing the full graph Laplacian during training. To address this limitation, numerous works emerged to enhance node classification based on GCN, such as methods, importance sampling node (FastGCN) (Chen et al., 2018), adaptive layer-wise sampling with skip connections (ASGCN) (Huang et al., 2018), adapting deep layers to GCN architectures (deepGCN) (Li et al., 2019), incorporating node-feature convolutional layers (NFC-GCN) (Zhang et al., 2022).

Two fundamental shortcomings of MPGNNs are the loss of structural information within node neighborhoods and the difficulty in capturing long-range dependencies (Khemani et al., 2024; Corso et al., 2024). To address these inherent issues, numerous studies conducted in the past few years have introduced various enhancements, including diverse aggregation schemes, such as skip connections (Chen et al., 2020), geometric methods (Pei et al., 2019), aiming to mitigate the risk of losing crucial information. Additionally, advancements like implicit hidden layers (Geng et al., 2021) and multiscale modeling (Liu et al., 2022) have been explored to augment the GNN's capabilities.

Another major issue with MPGNNs is their reliance on the homophily assumption, leading to poor performance in heterophilic networks (Zhu et al., 2020; Luan et al., 2024). Recent efforts have focused on designing GNNs that perform well in heterophilic settings (Lim et al., 2021a; Luan et al., 2022; Zheng et al., 2022; Xu et al., 2023; Zhao et al., 2024). While prior works analyze class-wise relations via neighborhood label distributions (Ma et al., 2021; Zhu et al., 2023) or class-compatibility matrices (Zhu et al., 2021), SCNode introduces a coordinate-based formulation that combines local structural distributions with global class-aware landmark distances, providing backbone-agnostic feature augmentations that capture long-range class relations.

In addition, graph kernels, such as Weisfeiler-Lehman (Shervashidze et al., 2011), shortest-path Borgwardt & Kriegel (2005), and graphlet kernels (Shervashidze et al., 2009), characterize similarity through handcrafted structural mappings (Vishwanathan et al., 2010). SCNode is related in spirit, but it differs in two important ways. It combines local structural distributions with global class-aware relational signals, providing richer contextual information than fixed kernel similarities. Moreover, SCNode is computationally more efficient, avoiding the quadratic complexity of kernel evaluations by generating embeddings in linear time that can be directly used within GNNs or graph transformers.

## 3 SCNode Framework

In the following discussion, we use the notation $\mathcal{G} = (\mathcal{V}, \mathbb{E}, \mathcal{W}, \mathcal{X})$ where $\mathcal{V} = \{v_i\}_{i=1}^m$ represents the vertices (nodes), $\mathbb{E} = \{e_{ij}\}$ represents edges, $\mathcal{W} = \{\omega_{ij} \subset \mathbb{R}^+\}$ represents edge weights, and $\mathcal{X} = \{\mathcal{X}_i\}_{i=1}^m \subset \mathbb{R}^n$ represents node features. If no node features are provided, $\mathcal{X} = \emptyset$. Consider a graph where nodes are categorized into $N$ classes. For each node $u$, we aim to construct an embedding $\vec{\gamma}(u)$ of dimension $q \times N$, where $q$ is determined by specifics of the graph such as its directedness, weighted nature, and the format of domain features.

Specifically, we generate *spatial embeddings* $\{\vec{\alpha}_1(u), \vec{\alpha}_2(u), \dots\}$, derived from local neighborhood information, and *contextual embeddings* $\{\vec{\beta}_1(u), \vec{\beta}_2(u), \dots\}$, derived from node properties specific to the domain. Each embedding, $\vec{\alpha}_i(u)$ and $\vec{\beta}_j(u)$, is $N$-dimensional, with one entry corresponding to each class (SCNode coordinates).

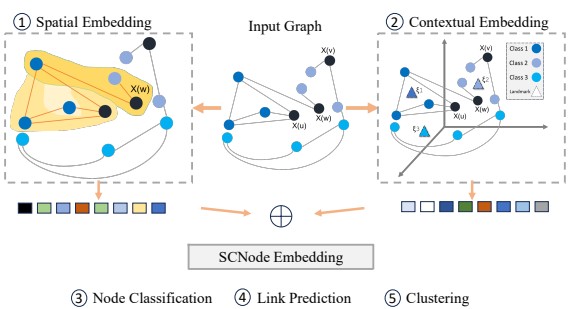

Figure 1: **SCNode framework**. Spatial and contextual embeddings are incorporated in class-aware contrast-based learning.

These embeddings are then concatenated to form the

final embedding $\vec{\gamma}(u)$ of size $q \times N$. For simplicity, we focus on applications in an inductive setting. In Appendix B.2, we discuss adaptations of our methods to accommodate transductive settings.

### 3.1 Spatial Node Embeddings

Spatial node embedding leverages the structure of a graph to measure the proximity of a node to various known classes within that graph. For each node $u$ in the graph $\mathcal{G}$ with vertices $\mathcal{V}$, the $k$-hop neighborhood of $u$, denoted as $\mathcal{N}_k(u)$, includes all vertices $v$ such that the shortest path distance $d(u, v)$ between $u$ and $v$ is at most $k$ hops: $\mathcal{N}_k(u) = \{v \in \mathcal{V} \mid d(u, v) \leq k\}$. Here, $d(u, v)$ represents the shortest path length between $u$ and $v$. If no path exists, then $d(u, v) = \infty$. In the case of a directed graph $\mathcal{G}$, edge directions are disregarded when calculating distances.

Assume that there are $N$ classes of nodes, represented as $\mathcal{C}_1, \mathcal{C}_2, \ldots, \mathcal{C}_N$. Let $\mathcal{V}_{\text{train}}$ be the set of nodes with known labels in the training dataset. The feature vector $\vec{\alpha}_k(u)$ for node $u$ is initialized as the distribution of class occurrences: $\vec{\alpha}_k(u) = [a_{k1}, a_{k2}, \ldots, a_{kN}]$ where $a_{kj}$ counts how many neighbors of $u$ belong to class $\mathcal{C}_j$ within the k-hop neighborhood intersecting with $\mathcal{V}_{\text{train}}$: $a_{kj} = |\{v \in \mathcal{C}_j \mid v \in \mathcal{N}_k(u) \cap \mathcal{V}_{\text{train}}\}|$. Consider the toy example of Figure 2(a), where the (k=1)-hop neighborhood of node $u$ contains node $z$ from class 1 and node $y$ from class 2. As a result, $\vec{\alpha}_1(u) = [1\ 1\ 0]$. We extend the spatial embeddings to directed and weighted graph cases by incorporating edge directions and weights in additional dimensions (see Appendix Section B.3) for the definitions.

### 3.2 Contextual Node Embeddings

In this section, we focus on extracting the most relevant information from the node attribute space by utilizing class-level reference points. Instead of relying solely on local neighborhood information, we assess each node's position relative to a set of representative class embeddings, or class landmarks, defined in the attribute space. These landmarks serve as stable points that reflect the typical feature profiles of different classes.

To compute contextual embeddings, we measure the similarity or distance between a node's attribute vector, $\mathcal{X}_u$, and these class landmarks. This approach enables the model to capture how closely a node aligns with each class, regardless of its immediate neighbors.

We use the term *contextual node embeddings* to emphasize the integration of node attributes with relational and structural information from the graph. Unlike traditional *attribute embeddings*, which focus solely on embedding node characteristics, contextual embeddings account for both the intrinsic properties of a node and its position within the graph structure. This approach is especially important in scenarios where node features and class behavior are intertwined, such as when the topic of an article influences its citation links. By combining attribute and relational context, our embeddings provide a richer representation of the graph, enabling improved performance in classification tasks.

In the following, we define class representatives (landmarks) in the node attribute space for each class. Let $\{\mathcal{C}_j\}_{j=1}^N$ denote the set of node classes. For each class $\mathcal{C}_j$, we identify the cluster of points

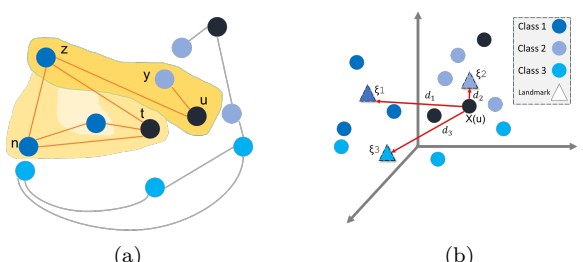

(a)        (b)

Figure 2: Computing spatial (a) and contextual (b) embeddings for node $u$ on the graph.

$\mathcal{Z}_j = \{\mathbf{X}(u) \mid u \in \mathcal{C}_j\}$ within the node attribute space $\mathbb{R}^n$, and establish a representative landmark $\xi_j \in \mathbb{R}^n$ for this cluster.

**Definition 3.1** (Class Landmark). Given a node class $\mathcal{C}_j$ and corresponding cluster of node embeddings $\mathcal{Z}_j$, the landmark $\xi_j$ is computed as the centroid of the points in $\mathcal{Z}_j$, normalized by the number of nodes in $\mathcal{C}_j$:

$$\xi_j = \frac{1}{|\mathcal{C}_j|} \sum_{u \in \mathcal{C}_j} \mathcal{X}(u)$$

**Definition 3.2** (Distances to Class Landmarks). Let $\mathbf{d}$ be a metric that measures the distance in $\mathbb{R}^n$. For a node $u$ with an embedding $\mathcal{X}(u)$, the distance to the class landmark $\xi_j$ is defined as: $d_j = \mathbf{d}(\mathcal{X}(u), \xi_j)$ for each class $\mathcal{C}_j$. These distances $d_j$ help in assessing how similar or distinct the node is from each class represented by the landmarks.

**Definition 3.3** (Contextual Embedding). Given a node $u$ and a set of class landmarks $\{\xi_j\} \subseteq \mathbb{R}^n$, one for each class $\mathcal{C}_j$, the contextual embedding of node $u$ is defined by a vector of distances from the node's embedding to each class landmark:
$$\vec{\beta}(u) = [d_1, d_2, d_3, \ldots, d_N]$$

where $d_j = \mathbf{d}(\mathcal{X}(u), \xi_j)$ for each class $\mathcal{C}_j$ (see Figure 2(b)). This vector $\vec{\beta}(u)$ encapsulates the node's position relative to each class within the attribute space.

**Class Landmarks.** To extract more detailed information about the domain attributes of nodes for each class, multiple landmarks can be defined. In many cases, the node attribute space lacks an inherent metric, and using different metrics can result in diverse landmarks and contextual embeddings. For example, the first landmark $\xi_j^1 \in \mathbb{R}^n$ may represent the center of the cluster $\mathcal{Z}_j$ using the Euclidean metric, while a second landmark $\xi_j^2 \in \mathbb{R}^n$ for the same class can be based on Jaccard similarity, capturing the most frequent attributes within $\mathcal{Z}_j$ (see Appendix C).

For each type of landmark, a corresponding distance or similarity measure $\mathbf{d}^k(.,.)$ is defined, such as Euclidean or cosine distance for real-valued attribute vectors, or Jaccard similarity for categorical attributes (Niwattanakul et al., 2013).

By leveraging different metrics and their corresponding landmarks, we compute an $N$-dimensional vector

$$\vec{\beta}_k(u) = [d_{k1}, d_{k2}, d_{k3}, \ldots, d_{kN}]$$

where $d_{kj} = \mathbf{d}^k(\mathbf{X}(u), \xi_j^k)$ represents the distance (or similarity) between node embeddings $\mathbf{X}(u)$ and the landmark $\xi_j^k$ of class $\mathcal{C}_j$.

### 3.3 SCNode Embedding

We may expand spatial neighborhoods and extend landmark sets arbitrarily. However, for exposition purposes, we will define SCNode embeddings over a directed graph where each node's neighborhood is considered up to two hops and each class $\mathcal{C}_j$ has two landmarks. We define the SCNode Embedding of a node $u$ by concatenating spatial and contextual embeddings. Specifically, we consider:

• *Spatial embeddings* from incoming and outgoing 1-hop neighborhoods ($\vec{\alpha}_{1i}(u)$ and $\vec{\alpha}_{1o}(u)$) and the 2-hop neighborhood ($\vec{\alpha}_2(u)$).

• *Contextual embeddings* based on distances to two class landmarks ($\vec{\beta}_1(u)$ and $\vec{\beta}_2(u)$), representing domain-specific characteristics.

**Definition 3.4** (SCNode Embedding)**.** The final embedding $\vec{\gamma}(u)$ for node $u$ is a concatenated vector of these embeddings as shown in the right. For example, if the graph is directed, and we want to utilize two landmark types, $\{\xi_j^1\}$ and $\{\xi_j^2\}$, the resulting SCNode vector will be in a $5 \cdot N$-dimensional vector where $N$ is the number of classes. For clarity, we represent $\vec{\gamma}(u)$ in a 2D format ($m \times N$), where each column corresponds to one class and each row represents one type of *SCNode* vector.

$$
\vec{\gamma}(u) = \begin{bmatrix} \longleftarrow & \vec{\alpha}_{1i}(u) & \longrightarrow \\ \longleftarrow & \vec{\alpha}_{1o}(u) & \longrightarrow \\ \longleftarrow & \vec{\alpha}_{2}(u) & \longrightarrow \\ \longleftarrow & \vec{\beta}_{1}(u) & \longrightarrow \\ \longleftarrow & \vec{\beta}_{2}(u) & \longrightarrow \end{bmatrix}
\begin{matrix} \text{Spatial} & \text{Incoming 1-ngbd} \\ \text{Spatial} & \text{Outgoing 1-ngbd} \\ \text{Spatial} & \text{2-ngbd} \\ \text{Contextual} & \{d(\mathbf{X}_u, \xi_j^1)\} \\ \text{Contextual} & \{d(\mathbf{X}_u, \xi_j^2)\} \end{matrix}
$$

**Relation to label-structure methods.** SCNode augments each node with two complementary signals: contextual coordinates given by distances to class landmarks in the embedding space, and spatial class-distribution histograms over k-hop neighborhoods. This encodes label structure over nodes without modifying the graph or introducing virtual nodes. In contrast, techniques such as LEGNN (Yu et al., 2022b) and GraphGPS (Rampášek et al., 2022) inject graph-level signals or virtual nodes derived from class labels. Conceptually, the class landmarks play the role of prototypes: distances to class-conditioned centroids provide barycentric coordinates in label space (Snell et al., 2017), but unlike prototypical networks (Hou & Sato, 2022), our prototypes are computed from the training graph and kept fixed during message passing, which decouples metric learning from the backbone and enables transductive inference. Our approach is backbone agnostic and operates at the feature level, so it pairs naturally with standard GNNs and graph transformers.

### 3.4 Homophily and SCNode

In this section, we demonstrate the effectiveness of the SCNode framework in analyzing homophily behavior within the graph from multiple perspectives and gaining deeper insights into class interactions. In recent years, several metrics have been introduced to study the effect of homophily on graph representation learning (Lim et al., 2021b; Jin et al., 2022; Luan et al., 2022; Platonov et al., 2023a) (see overview in Appendix A.1). A widely used metric, the *node homophily ratio*, is defined as $\mathcal{H}_{\text{node}}(\mathcal{G}) = \frac{1}{|\mathcal{V}|} \sum_{v \in \mathcal{V}} \frac{\eta(v)}{deg(v)}$, where $\eta(v)$ represents the number of adjacent nodes to $v$ sharing the same class. A graph $\mathcal{G}$ is termed *homophilic* if $\mathcal{H}_{\text{node}}(\mathcal{G}) \geq 0.5$, and *heterophilic* otherwise.

Although homophily measures similarity across an entire graph, individual groups within a graph may display different homophily behaviors. Our SCNode approach leverages class interactions and introduces a class-aware homophily score through non-symmetric measures:

**Definition 3.5** (SCNode Homophily Matrices)**.** Let $(\mathcal{G}, \mathcal{V})$ be graph with node classes $\{\mathcal{C}_1, \mathcal{C}_2, \ldots, \mathcal{C}_N\}$. Let $\vec{\alpha}$ be a spatial or contextual embedding. We define the homophily rate between classes $i$ and $j$ as $h_{ij}^{\alpha} = \frac{1}{|\mathcal{C}_i|} \sum_{v \in \mathcal{C}_i} \frac{\alpha_v^j}{|\alpha_v|}$ where $\alpha_v^j$ is the $j^{th}$ entry of $\vec{\alpha}_v$. Considering pairwise homophily rates of all classes, we create the $N \times N$ matrix $\mathbf{M} = [h_{ij}^{\alpha}]$ as the $\alpha$ *Homophily Matrix* of $\mathcal{G}$.

Table 9 in the Appendix presents examples of SCN homophily matrices. These matrices offer detailed insights into intra-class (homophily) and inter-class (heterophily) interactions across spatial and contextual contexts, with the diagonal elements representing the likelihood of nodes connecting within their own class.

**Definition 3.6** ($\alpha$ Homophily Ratio)**.** For a given $\alpha$ Homophily Matrix $\mathcal{M}^{\alpha}$, $\alpha$ Homophily $\mathcal{H}_{scn}^{\alpha}$ is defined as the average of the diagonal elements. i.e, $\mathcal{H}_{scn}^{\alpha} = \frac{1}{N} \sum_{i=1}^{N} h_{ii}$.

For example, the Spatial-1 Homophily ratio reveals a node's likelihood to connect with nodes of the same class within its immediate neighborhood. Homophily matrices not only introduce new ways to measure homophily but also relate to various existing homophily metrics:

**Theorem 3.7.** *For a given $\mathcal{G} = (\mathcal{V}, \mathbb{E})$, let $\vec{\alpha}_1(v)$ be SCN spatial-1 vector. Let $\widehat{\alpha}_1(v)$ be the vector where the entry corresponding to the class of $v$ is set to 0. Then, $1 - \mathcal{H}_{node}(\mathcal{G}) = \frac{1}{|\mathcal{V}|} \sum_{v \in \mathcal{V}} \frac{\|\widehat{\alpha}_1(v)\|_1}{\|\vec{\alpha}_1(v)\|_1}$.*

The proof of the theorem and further discussions on how our SCNode homophily matrices relate to other forms of homophily metrics are detailed in the Appendix A.

## 4 Experiments

We evaluate SCNode in two tasks: node classification and link prediction. We share our Python implementation at `https://github.com/joshem163/SCNode`

**Datasets.** We use three widely-used homophilic datasets, which are citation networks, CORA, CITE-SEER, and PUBMED (Sen et al., 2008), two Open Graph Benchmark (OGB) datasets: OGBN-ARXIV and OGBN-MAG (Hu et al., 2020), and ten heterophilic datasets, including TEXAS, CORNELL, WISCON-SIN, and CHAMELEON (Pei et al., 2019), as well as filtered versions of SQUIRREL and CHAMELEON, AMAZON-RATINGS, TOLOKER, and QUESTIONS (Platonov et al., 2023b). The datasets are described in Appendix C and their statistics are given in Table 1.

Table 1: Benchmark datasets for node classification.

| Datasets | Nodes | Edges | Classes | Features | Tr/Val/Test (%) | Hom. |
|---|---|---|---|---|---|---|
| CORA | 2,708 | 5,429 | 7 | 1,433 | 48/32/20 | 0.83 |
| CITESEER | 3,312 | 4,732 | 6 | 3,703 | 48/32/20 | 0.72 |
| PUBMED | 19,717 | 44,338 | 3 | 500 | 48/32/20 | 0.79 |
| OGBN-ARXIV | 169,343 | 1,166,243 | 40 | 128 | OGB | 0.65 |
| OGBN-MAG | 1,939,743 | 21,111,007 | 349 | 128 | OGB | 0.30 |
| TEXAS | 183 | 309 | 5 | 1,703 | 48/32/20 | 0.10 |
| CORNELL | 183 | 295 | 5 | 1,703 | 48/32/20 | 0.39 |
| WISCONSIN | 251 | 499 | 5 | 1,703 | 48/32/20 | 0.15 |
| CHAMELEON | 2,277 | 36,101 | 5 | 2,325 | 48/32/20 | 0.25 |
| SQUIRREL* | 2,223 | 46,998 | 5 | 2,089 | 48/32/20 | 0.19 |
| CHAMELEON* | 890 | 8,854 | 5 | 2,325 | 48/32/20 | 0.24 |
| ROMAN-EMPIRE | 22,662 | 32,927 | 18 | 300 | 50/25/25 | 0.05 |
| AMAZON-RATINGS | 24,492 | 93,050 | 5 | 300 | 50/25/25 | 0.37 |
| TOLOKERS | 11,758 | 5,19,000 | 2 | 10 | 50/25/25 | 0.63 |
| QUESTIONS | 48,921 | 153,540 | 2 | 301 | 50/25/25 | 0.89 |

\* denotes the filtered version of the dataset.

**Experimental Setup.** We evaluate SCNode on three citation networks and four heterophilic graph datasets, following the experimental protocol established by Bodnar et al. (Bodnar et al., 2022). For these datasets, we randomly split the nodes of each class into 48% for training, 32% for validation, and 20% for testing, and report the average accuracy over 10 independent runs. For the OGB datasets, OGBN-ARXIV and OGBN-MAG, we use the official train/validation/test splits provided by the Open Graph Benchmark (OGB) (Hu et al., 2020), and report the classification accuracy accordingly. For the remaining datasets, we adopt the same splitting strategy as outlined in (Platonov et al., 2023b). We give the details of SCNode embeddings for each dataset in Appendix C. The dimensions of the embeddings used for each dataset are given in Table 11.

For classification, we employ a Multi-Layer Perceptron (MLP) with a single hidden layer consisting of 100 neurons. We use the ReLU activation function, a learning rate of 0.001, and the Adam optimizer for training up to 500 epochs. To mitigate overfitting, we apply L2 regularization with a weight decay of 0.0001.

All non-OGB experiments were conducted on a local machine equipped with an Apple M2 chip (8-core CPU, 10-core GPU, 6-core Neural Engine) and 16 GB of RAM. OGB experiments were run on Google Colab, using a system with an Intel(R) 2.20GHz CPU, NVIDIA V100 GPU, and 25.5 GB of RAM.

**Runtime.** SCNode is computationally efficient. SCNode requires approximately 4 hours for OGBN-ARXIV and 10 hours for OGB-MAG to create all embeddings, and about 10 minutes to train for all OGB datasets. End-to-end, our model processes three citation network datasets and three WebKb datasets, including the CHAMELEON dataset, in under a minute. PUBMED takes about 20 minutes. For comparison,

Table 2: **Node Classification Performance.** Accuracy results for node classification tasks. Baseline results up to Gen-NSD are sourced from (Bodnar et al., 2022; Li et al., 2022), while the remaining results are taken from their respective original papers. The best results are shown in **bold**, while the second-best results are underlined. The last column reports the average performance gap (%) relative to the best result across all datasets.

| Dataset | CORA | CITESEER | PUBMED | TEXAS | CORNELL | WISC. | CHAM. | Avg. Gap |
|---|---|---|---|---|---|---|---|---|
| Node Homophily | 0.83 | 0.72 | 0.79 | 0.10 | 0.39 | 0.15 | 0.25 | % |
| GCN (Kipf & Welling, 2017) | $86.14_{\pm1.10}$ | $75.51_{\pm1.28}$ | $87.22_{\pm0.37}$ | $56.22_{\pm5.81}$ | $60.54_{\pm5.30}$ | $51.96_{\pm5.17}$ | $65.94_{\pm3.23}$ | 19.9 |
| GraphSAGE (Hamilton et al., 2017) | $86.26_{\pm1.54}$ | $76.04_{\pm1.30}$ | $88.45_{\pm0.50}$ | $75.95_{\pm5.01}$ | $75.95_{\pm5.01}$ | $81.18_{\pm5.56}$ | $58.73_{\pm1.68}$ | 10.4 |
| GAT (Veličković et al., 2018a) | $85.03_{\pm1.61}$ | $76.55_{\pm1.23}$ | $87.30_{\pm1.10}$ | $54.32_{\pm6.30}$ | $61.89_{\pm5.05}$ | $49.41_{\pm4.09}$ | $60.26_{\pm2.50}$ | 20.1 |
| Geom-GCN (Pei et al., 2019) | $85.35_{\pm1.57}$ | $\mathbf{78.02_{\pm1.15}}$ | $89.95_{\pm0.47}$ | $66.76_{\pm2.72}$ | $60.54_{\pm3.67}$ | $64.51_{\pm3.66}$ | $60.00_{\pm2.81}$ | 15.8 |
| H2GCN (Zhu et al., 2020) | $87.87_{\pm1.20}$ | $77.11_{\pm1.57}$ | $89.49_{\pm0.38}$ | $84.86_{\pm7.23}$ | $82.70_{\pm5.28}$ | $87.65_{\pm4.98}$ | $60.11_{\pm2.15}$ | 8.9 |
| GPRGCN (Chien et al., 2020) | $87.95_{\pm1.18}$ | $77.13_{\pm1.67}$ | $87.54_{\pm0.38}$ | $81.35_{\pm5.32}$ | $78.11_{\pm6.55}$ | $82.55_{\pm6.23}$ | $46.58_{\pm1.71}$ | 9.7 |
| GCNII (Chen et al., 2020) | $88.37_{\pm1.25}$ | $77.33_{\pm1.48}$ | $90.15_{\pm0.43}$ | $77.57_{\pm3.83}$ | $77.86_{\pm3.79}$ | $80.39_{\pm3.40}$ | $63.86_{\pm3.04}$ | 8.2 |
| WRGAT (Suresh et al., 2021) | $88.20_{\pm2.26}$ | $76.81_{\pm1.89}$ | $88.52_{\pm0.92}$ | $83.62_{\pm5.50}$ | $81.62_{\pm3.90}$ | $86.98_{\pm3.78}$ | $65.24_{\pm0.87}$ | 8.1 |
| LINKX (Lim et al., 2021a) | $84.64_{\pm1.13}$ | $73.19_{\pm0.99}$ | $87.86_{\pm0.77}$ | $74.60_{\pm8.37}$ | $77.84_{\pm5.81}$ | $75.49_{\pm5.72}$ | $68.42_{\pm1.38}$ | 10.6 |
| NLGAT (Liu et al., 2021) | $\underline{88.50_{\pm1.80}}$ | $76.20_{\pm1.60}$ | $88.20_{\pm0.30}$ | $62.60_{\pm7.10}$ | $54.70_{\pm7.60}$ | $56.90_{\pm7.30}$ | $65.70_{\pm1.40}$ | 17.5 |
| GloGNN++ (Li et al., 2022) | $88.33_{\pm1.09}$ | $77.22_{\pm1.78}$ | $89.24_{\pm0.39}$ | $84.05_{\pm4.90}$ | $85.95_{\pm5.10}$ | $88.04_{\pm3.22}$ | $71.21_{\pm1.84}$ | 4.5 |
| GGCN (Yan et al., 2022) | $87.95_{\pm1.05}$ | $77.14_{\pm1.45}$ | $89.15_{\pm0.37}$ | $84.86_{\pm4.55}$ | $85.68_{\pm6.63}$ | $86.86_{\pm3.29}$ | $71.14_{\pm1.84}$ | 4.7 |
| Gen-NSD (Bodnar et al., 2022) | $87.30_{\pm1.15}$ | $76.32_{\pm1.65}$ | $89.33_{\pm0.35}$ | $82.97_{\pm5.13}$ | $85.68_{\pm6.51}$ | $89.21_{\pm3.84}$ | $67.93_{\pm1.58}$ | 5.3 |
| ACM-GCN (Luan et al., 2022) | $87.91_{\pm0.95}$ | $77.32_{\pm1.70}$ | $90.00_{\pm0.52}$ | $87.84_{\pm4.40}$ | $85.14_{\pm6.07}$ | $88.43_{\pm3.22}$ | $69.14_{\pm1.91}$ | 4.3 |
| LRGNN (Liang et al., 2023) | $88.30_{\pm0.90}$ | $77.50_{\pm1.30}$ | $\underline{90.20_{\pm0.60}}$ | $90.30_{\pm4.50}$ | $86.50_{\pm5.70}$ | $88.20_{\pm3.50}$ | $\underline{79.16_{\pm2.05}}$ | 2.2 |
| Ordered-GNN (Song et al., 2023) | $88.37_{\pm0.75}$ | $77.31_{\pm1.73}$ | $90.15_{\pm0.38}$ | $86.22_{\pm4.12}$ | $\underline{87.03_{\pm4.73}}$ | $88.04_{\pm3.63}$ | $72.28_{\pm2.29}$ | 3.7 |
| TEDGCN (Yan et al., 2024) | $87.90_{\pm1.31}$ | $77.81_{\pm1.72}$ | $84.84_{\pm1.21}$ | $\underline{91.41_{\pm3.62}}$ | $86.53_{\pm3.80}$ | $\underline{91.42_{\pm4.22}}$ | $67.33_{\pm1.71}$ | 4.1 |
| FGSAM-SAGE (Luo et al., 2024a) | $88.36_{\pm1.51}$ | $77.13_{\pm0.69}$ | $89.75_{\pm0.49}$ | $81.35_{\pm5.10}$ | $82.43_{\pm3.83}$ | $86.47_{\pm4.31}$ | $51.34_{\pm2.96}$ | 8.4 |
| **SCNode** | $\mathbf{88.65_{\pm1.25}}$ | $\underline{77.83_{\pm1.60}}$ | $\mathbf{90.53_{\pm0.61}}$ | $\mathbf{94.59_{\pm5.25}}$ | $\mathbf{88.09_{\pm1.91}}$ | $\mathbf{92.01_{\pm4.06}}$ | $\mathbf{84.08_{\pm1.55}}$ | 0.1 |

RevGNN-Deep requires 13.5 days and RevGNN-Wide takes 17.1 days to train for 2000 epochs on a single NVIDIA V100 for the OGB datasets (Li et al., 2021).

## 4.1 Node Classification Results

**Baselines.** We compare SCNode against a range of state-of-the-art models( Tables 2 to 4), including three classical approaches: GCN (Kipf & Welling, 2017), GraphSAGE (Hamilton et al., 2017), and GAT (Veličković et al., 2018a), as well as a graph transformer model. All baselines in Tables 2 to 4 are evaluated in the transductive setting, except for GraphSAGE, which follows the inductive setting.

**Performance.** We present the node classification results for benchmark homophilic and heterophilic datasets in Tables 2 and 3 and for OGB datasets in Table 4. On the two homophilic benchmarks, SCNODE achieves state-of-the-art performance, consistently outperforming prior methods. Even more strikingly, on the four heterophilic graphs, SCNODE delivers dramatic gains, surpassing the previous best by 2 – 4 points. For the remaining five heterophilic datasets, SCNODE achieves the best results on two and demonstrates competitive performance on the others, as shown in Table 3. This uniform superiority across low- and high-homophily settings underscores SCNODE's ability to adaptively integrate both local feature similarity and global topological cues, yielding a single architecture that excels regardless of underlying graph structure. Moreover, on the OGB benchmarks, SCNODE achieves results within 3-4 % of the state-of-the-art, despite relying on a far more compact architecture than competing deep-learning models.

**SCN-GNNs: GNNs with SCNode Embeddings.** To evaluate the integration of SCNode with GNNs as plug-and-play components, *we replaced the initial node embeddings in GNNs with SCN embeddings* and assessed their effectiveness. We tested three classical GNN models (GCN, GraphSAGE, and GAT), along with the recent LINKX model. A two-layer GNN framework was implemented using the Adam optimizer with an initial learning rate of 0.001, a dropout rate of 0.5, a weight decay of 5e-4, and 32 hidden channels. The results, shown in Table 5 and Figures 3 and 4, indicate significant performance improvements when using SCN embeddings. These embeddings accelerate convergence and maintain a consistent accuracy advantage over the vanilla models. Notably, combining SCNode with the LINKX model yields the best results across all baselines for both datasets, showcasing the potential of SCNode to enhance GNN performance. This improvement is attributed to the effective integration of spatial and contextual information provided by

Table 3: **More Heterophilic Datasets.** Node classification results of our models compared to GNN and graph transformer baselines on heterophilic benchmarks. The last column reports the average gap to the best-performing model in each dataset. Baseline models are sourced from (Platonov et al., 2023b; Yadati, 2025; Luo et al., 2024b)

| Model | Squirrel* | Chameleon* | Amazon | Tolokers | Questions | Avg. Gap |
|---|---|---|---|---|---|---|
| GCN (Kipf & Welling, 2017) | $34.50_{\pm1.61}$ | $38.53_{\pm2.23}$ | $47.94_{\pm0.62}$ | $83.64_{\pm0.67}$ | $63.04_{\pm1.61}$ | 8.58 |
| GraphSAGE (Hamilton et al., 2017) | $35.19_{\pm2.49}$ | $36.00_{\pm2.76}$ | $53.44_{\pm0.48}$ | $82.43_{\pm0.44}$ | $75.27_{\pm1.20}$ | 5.64 |
| GAT (Veličković et al., 2018a) | $34.41_{\pm0.96}$ | $38.79_{\pm3.07}$ | $49.25_{\pm0.73}$ | $44.98_{\pm1.96}$ | $73.42_{\pm1.63}$ | 13.94 |
| H2GCN (Zhu et al., 2020) | $35.10_{\pm1.15}$ | $26.75_{\pm3.64}$ | $36.47_{\pm0.23}$ | $73.35_{\pm1.01}$ | $63.59_{\pm1.46}$ | 15.05 |
| GPRGNN (Chien et al., 2020) | $38.95_{\pm1.99}$ | $39.93_{\pm3.30}$ | $44.88_{\pm0.82}$ | $72.94_{\pm0.97}$ | $55.48_{\pm0.91}$ | 11.67 |
| CPGNN (Zhu et al., 2021) | $30.04_{\pm2.03}$ | $33.00_{\pm3.15}$ | $39.79_{\pm0.77}$ | $73.36_{\pm1.01}$ | $65.96_{\pm1.95}$ | 13.68 |
| FSGNN (Maurya et al., 2022) | $35.92_{\pm1.32}$ | $35.60_{\pm2.97}$ | $52.74_{\pm0.83}$ | $82.76_{\pm0.61}$ | $78.86_{\pm0.92}$ | 4.93 |
| GloGNN (Li et al., 2022) | $35.11_{\pm1.24}$ | $25.90_{\pm1.82}$ | $36.89_{\pm0.14}$ | $73.39_{\pm1.17}$ | $65.74_{\pm1.19}$ | 14.70 |
| LRGNN (Liang et al., 2023) | $39.51_{\pm2.12}$ | $41.24_{\pm2.95}$ | $42.23_{\pm4.85}$ | - | $66.41_{\pm1.75}$ | 9.08 |
| NodeFormer (Wu et al., 2022) | $38.52_{\pm1.57}$ | $34.78_{\pm4.14}$ | $43.86_{\pm0.35}$ | $78.10_{\pm1.03}$ | $74.27_{\pm1.14}$ | 8.20 |
| GraphGPS (Rampášek et al., 2022) | $35.49_{\pm2.00}$ | $41.04_{\pm1.11}$ | $44.94_{\pm0.77}$ | $83.71_{\pm0.48}$ | $72.15_{\pm1.16}$ | 6.64 |
| SGFormer (Wu et al., 2023) | $35.81_{\pm2.02}$ | $42.54_{\pm3.58}$ | $48.18_{\pm0.71}$ | $83.33_{\pm0.68}$ | $73.05_{\pm1.16}$ | 5.52 |
| Polynormer (Deng et al., 2024) | $40.87_{\pm1.96}$ | $41.82_{\pm3.45}$ | $\mathbf{54.81_{\pm0.49}}$ | $\mathbf{84.83_{\pm0.72}}$ | $\mathbf{78.92_{\pm0.89}}$ | **1.86** |
| **SCNode** | $\mathbf{45.85_{\pm2.36}}$ | $\mathbf{46.12_{\pm3.21}}$ | $52.57_{\pm0.42}$ | $79.99_{\pm0.41}$ | $76.04_{\pm1.62}$ | 1.99 |

Table 4: Classification accuracy of baselines and our SCNode model on **OGBN datasets**.

| Model | ARXIV | MAG |
|---|---|---|
| GCN (Kipf & Welling, 2017) | 71.74 | 34.87 |
| GSAGE (Hamilton et al., 2017) | 71.49 | 37.04 |
| GAT (Veličković et al., 2018a) | 73.91 | 37.67 |
| DeepGCN (Li et al., 2019) | 72.32 | – |
| DAGNN (Liu et al., 2020) | 72.09 | – |
| UniMP-v2 (Shi et al., 2020) | 73.92 | – |
| RevGAT (Li et al., 2021) | **74.26** | – |
| RGCN (Yu et al., 2022a) | – | 47.96 |
| HGT (Yu et al., 2022a) | – | 49.21 |
| R-HGNN (Yu et al., 2022a) | – | 52.04 |
| LEGNN (Yu et al., 2022b) | 73.71 | 53.78 |
| S3GCL (Wan et al., 2024) | 71.36 | – |
| LMSPS (Liu et al., 2024) | – | **54.83** |
| **SCNode** | 71.56 | 50.03 |

Table 5: Vanilla-GNN vs. SCNode+GNN (SCN-GNN) accuracy results for node classification.

| Dataset | Model | GNN | SCN-GNN | Imp.($\uparrow$) |
|---|---|---|---|---|
| **CORA** | GCN | $86.14_{\pm1.10}$ | $88.43_{\pm0.92}$ | **2.29** |
| | SAGE | $86.26_{\pm1.54}$ | $\mathbf{88.98_{\pm1.37}}$ | **2.72** |
| | GAT | $85.03_{\pm1.61}$ | $87.18_{\pm2.12}$ | **2.15** |
| | LINKX | $81.63_{\pm1.57}$ | $83.80_{\pm1.43}$ | **2.17** |
| **CITESEER** | GCN | $75.39_{\pm1.92}$ | $77.18_{\pm1.98}$ | **1.79** |
| | SAGE | $74.65_{\pm1.58}$ | $77.36_{\pm1.18}$ | **2.71** |
| | GAT | $74.85_{\pm1.46}$ | $\mathbf{77.47_{\pm1.59}}$ | **2.62** |
| | LINKX | $70.51_{\pm2.48}$ | $73.04_{\pm1.76}$ | **2.53** |
| **TEXAS** | GCN | $56.22_{\pm5.81}$ | $70.81_{\pm6.43}$ | **14.59** |
| | SAGE | $75.95_{\pm5.01}$ | $87.84_{\pm6.65}$ | **11.89** |
| | GAT | $54.32_{\pm6.30}$ | $62.16_{\pm5.70}$ | **7.84** |
| | LINKX | $74.60_{\pm8.37}$ | $\mathbf{93.78_{\pm4.04}}$ | **19.18** |
| **WISCONSIN** | GCN | $51.96_{\pm5.17}$ | $63.53_{\pm4.45}$ | **11.57** |
| | SAGE | $81.18_{\pm5.56}$ | $84.90_{\pm3.21}$ | **3.72** |
| | GAT | $49.41_{\pm4.09}$ | $55.29_{\pm6.64}$ | **5.88** |
| | LINKX | $75.49_{\pm5.72}$ | $\mathbf{91.18_{\pm1.39}}$ | **15.69** |

SCNode embeddings. The accuracy gap remains stable, further demonstrating their robustness. In future work, we aim to explore this integration more comprehensively to further enhance GNN performance.

## 4.2 Link Prediction Results

In this part, we show the utilization of SCNode Embeddings in the link prediction task. We utilize the common setting described in (Zhou et al., 2022) where the datasets were partitioned into training, validation, and test sets with a ratio of $85\%, 5\%$, and $10\%$, respectively.

The model architecture for the link prediction problem consists of three MLP layers. In this framework, for each considered node pair $(u, v)$, the normalized SCNode encodings are term-wise multiplied, $h_u \cdot h_v$, and feed into MLP. We configure the model with a hidden neuron dimension of 16, a learning rate of 0.001, and train it over 100 epochs with a batch size of 128.

Table 6 reports the prediction results. We report the baselines from (Fu et al., 2023) and (Zhou et al., 2022), which use our experiment settings. Out of three homophilic and three heterophilic datasets, SCNode outperforms existing models in three datasets and gets the second best result in one. **On the six datasets, SCNode reaches the highest mean AUC of 0.905.**

**Ablation Study.** To evaluate the efficacy of our feature vectors, we conducted ablation studies for the node classification task. We employed three submodels: utilizing (1) spatial embeddings, (2) contectual embeddings, and (3) both (SCNode embeddings). As given in Table 7, we observe that in the CORA,

Table 6: **Link Prediction Performances.** AUC results for Link Prediction. Baselines are reported from (Fu et al., 2023; Zhou et al., 2022; Wu et al., 2024). In the Overall column, we report mean AUC results across all datasets.

| Dataset | CORA | CITESEER | PUBMED | WISC. | CORNELL | TEXAS | Overall |
|---|---|---|---|---|---|---|---|
| Node2vec (Grover et al., 2016) | $0.856_{\pm 0.015}$ | $0.894_{\pm 0.014}$ | $0.919_{\pm 0.004}$ | – | – | – | – |
| GAE (Kipf & Welling, 2016) | $0.895_{\pm 0.165}$ | $0.887_{\pm 0.084}$ | $0.957_{\pm 0.012}$ | $0.689_{\pm 0.384}$ | $0.736_{\pm 1.090}$ | $0.753_{\pm 1.297}$ | 0.820 |
| VGAE (Kipf & Welling, 2016) | $0.852_{\pm 0.493}$ | $0.810_{\pm 0.339}$ | $0.929_{\pm 0.134}$ | $0.669_{\pm 0.866}$ | $0.783_{\pm 0.401}$ | $0.767_{\pm 0.557}$ | 0.802 |
| ARVGE (Pan et al., 2018) | $0.913_{\pm 0.079}$ | $0.878_{\pm 0.177}$ | $0.965_{\pm 0.015}$ | $0.711_{\pm 0.377}$ | $\underline{0.789}_{\pm 0.501}$ | $0.765_{\pm 0.468}$ | 0.837 |
| DGI (Veličković et al., 2018b) | $0.898_{\pm 0.080}$ | $0.955_{\pm 0.100}$ | $0.912_{\pm 0.060}$ | – | – | – | – |
| G-VAE (Grover et al., 2019) | $0.947_{\pm 0.011}$ | $0.973_{\pm 0.006}$ | $0.974_{\pm 0.004}$ | – | – | – | – |
| GNAE (Ahn & Kim, 2021) | $0.941_{\pm 0.063}$ | $0.969_{\pm 0.022}$ | $0.954_{\pm 0.019}$ | $0.782_{\pm 0.829}$ | $0.729_{\pm 1.083}$ | $0.751_{\pm 1.067}$ | 0.854 |
| VGNAE (Ahn & Kim, 2021) | $0.892_{\pm 0.067}$ | $0.955_{\pm 0.055}$ | $0.897_{\pm 0.040}$ | $0.703_{\pm 0.120}$ | $0.733_{\pm 0.573}$ | $0.789_{\pm 0.302}$ | 0.828 |
| GIC (Mavromatis et al., 2021) | $0.935_{\pm 0.060}$ | $0.970_{\pm 0.050}$ | $0.937_{\pm 0.030}$ | – | – | – | – |
| LINKX (Lim et al., 2021a) | $0.934_{\pm 0.030}$ | $0.935_{\pm 0.050}$ | – | $0.801_{\pm 0.380}$ | – | $0.758_{\pm 0.470}$ | 0.857 |
| DisenLink (Zhou et al., 2022) | $\mathbf{0.971}_{\pm \mathbf{0.040}}$ | $\mathbf{0.983}_{\pm \mathbf{0.030}}$ | – | $\underline{0.844}_{\pm 0.190}$ | – | $0.807_{\pm 0.400}$ | $\underline{0.901}$ |
| DGAE (Fu et al., 2023) | $0.958_{\pm 0.044}$ | $0.972_{\pm 0.034}$ | $\underline{0.978}_{\pm 0.012}$ | $0.757_{\pm 0.586}$ | $0.681_{\pm 1.207}$ | $0.683_{\pm 1.279}$ | 0.838 |
| VDGAE (Fu et al., 2023) | $0.959_{\pm 0.042}$ | $\underline{0.978}_{\pm 0.030}$ | $0.970_{\pm 0.012}$ | $0.850_{\pm 0.478}$ | $0.761_{\pm 0.475}$ | $\underline{0.813}_{\pm 0.849}$ | 0.889 |
| HAGNN (Wu et al., 2024) | $0.936_{\pm 0.005}$ | $0.924_{\pm 0.008}$ | $0.967_{\pm 0.003}$ | $\mathbf{0.858}_{\pm \mathbf{0.048}}$ | $0.770_{\pm 0.036}$ | $0.795_{\pm 0.052}$ | 0.875 |
| **SCNode** | $\underline{0.967}_{\pm 0.047}$ | $0.952_{\pm 0.059}$ | $\mathbf{0.980}_{\pm \mathbf{0.012}}$ | $0.796_{\pm 0.338}$ | $\mathbf{0.814}_{\pm \mathbf{0.413}}$ | $\mathbf{0.831}_{\pm \mathbf{0.393}}$ | $\mathbf{0.905}$ |

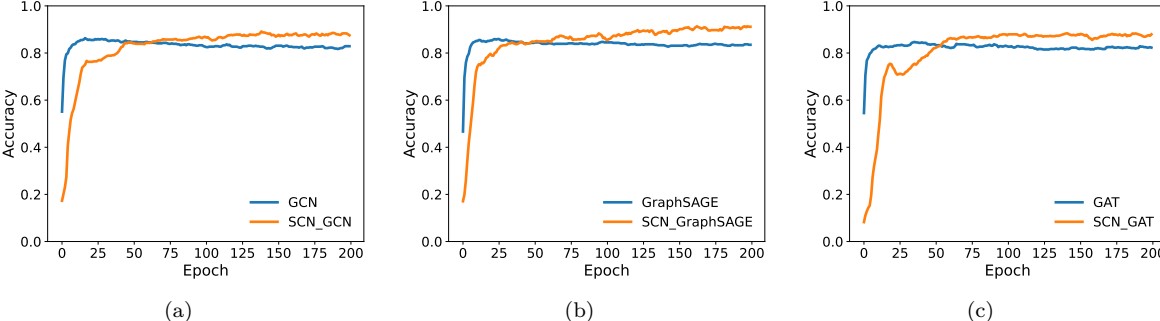

Figure 3: Performance comparison of three GNN models (GCN, GSAGE, GAT) starting with the original node embeddings (blue) and SCNode node embeddings (orange) on the CORA dataset.

CHAMELEON, and PUBMED datasets, using only spatial or domain feature embeddings individually yields satisfactory performance. However, their combination significantly enhances performance in most cases. CHAMELEON (+5.27) experiences a significant increase in accuracy in the combined setting. We note that when the ablation study has a large accuracy gap between the spatial and domain-only models for a dataset (e.g., in TEXAS), the accuracies of the SOTA models in Table 2 show huge accuracy deviations for the dataset as well (e.g., TEXAS accuracies range from 54.32 to 90.30). A possible explanation is that models might individually capture either spatial or contextual information. Hence, they may be unable to combine these two sets of features to counterbalance the insufficient information present in one of them, leading to diminished accuracy scores. In contrast, the ablation study offers evidence that the SCNode approach is resilient to this limitation and experiences accuracy gain (e.g., 91.35 → 94.59 for TEXAS in Table 7). We present further ablation studies on the effect of neighborhood size choice on spatial embeddings, and similarity metric choice on contextual embeddings in Appendix B.4.

Table 7: **Spatial vs. Contextual.** Accuracy results of our model considering different feature subsets.

| Features | CORA | CITESEER | PUBMED | TEXAS | CORNELL | WISC. | CHAM. |
|---|---|---|---|---|---|---|---|
| Spatial Only | $84.61_{\pm 1.20}$ | $71.82_{\pm 1.95}$ | $85.86_{\pm 0.32}$ | $67.29_{\pm 6.29}$ | $49.45_{\pm 4.23}$ | $57.84_{\pm 6.14}$ | $63.70_{\pm 3.05}$ |
| Context Only | $73.98_{\pm 2.56}$ | $67.42_{\pm 0.85}$ | $89.70_{\pm 0.50}$ | $91.35_{\pm 4.37}$ | $\mathbf{92.71}_{\pm \mathbf{4.23}}$ | $\mathbf{94.71}_{\pm \mathbf{2.62}}$ | $78.81_{\pm 1.33}$ |
| Both (SCNode) | $\mathbf{88.65}_{\pm \mathbf{1.25}}$ | $\mathbf{77.83}_{\pm \mathbf{1.60}}$ | $\mathbf{90.53}_{\pm \mathbf{0.61}}$ | $\mathbf{94.59}_{\pm \mathbf{5.25}}$ | $88.09_{\pm 1.91}$ | $92.01_{\pm 4.06}$ | $\mathbf{84.08}_{\pm \mathbf{1.55}}$ |

**Effectiveness in smaller training Settings.** Our model leverages both spatial and contextual encodings, where contextual embeddings are derived from class cluster centroids as landmarks, and spatial encodings capture class distributions within k-hop neighborhoods. For these embeddings to be ef-

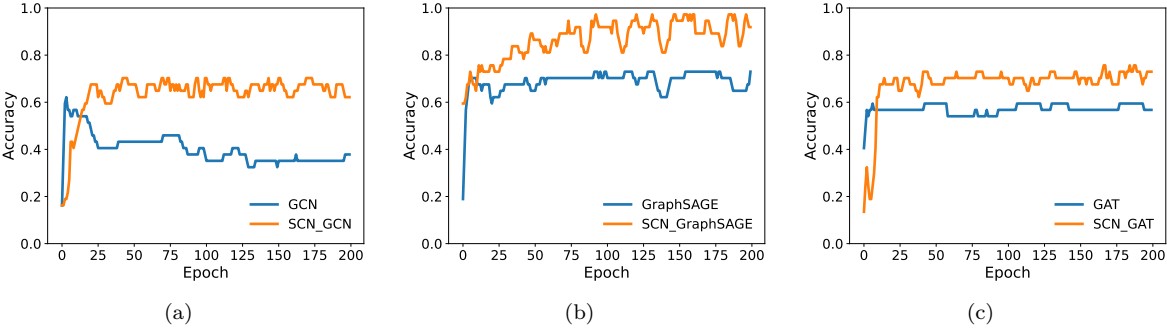

Figure 4: Performance comparison of three GNN models (GCN, GraphSAGE, GAT) starting with the original node embeddings (blue) and SCNode node embeddings (orange) on the TEXAS dataset.

fective, two conditions are particularly important: reliable landmarks and a balanced class distribution. To evaluate the robustness of SCNode under limited supervision, we fix 20% of the nodes as the test set and vary the size of the training set from 5% to 75% across three benchmark datasets: Cora, Citeseer, and PubMed (see Figure 5). The results show that SCNode achieves stable performance across all datasets once at least 50% of the nodes are included in the training set. It is also worth noting that on PubMed, our model performs noticeably well even under low training ratios. This is because, despite only using 5% of nodes for training, the dataset still provides nearly 1,000 labeled nodes across only three classes. This ensures a sufficient number of representatives per class, enabling SCNode to learn effective contextual embeddings. Consequently, SCNode produces stronger node representations when enough class representatives are available, making the embeddings more effective for supervised learning.

**Limitations.** A limitation of SCNode lies in its reliance on the quality and relevance of the landmarks used to define the contextual embeddings. If these landmarks fail to adequately represent the underlying data distributions, the model's performance may suffer. However, this issue could be mitigated by leveraging multiple landmarks or employing contrastive learning in an unsupervised setting to optimize landmark selection. We aim to address this limitation in future work.

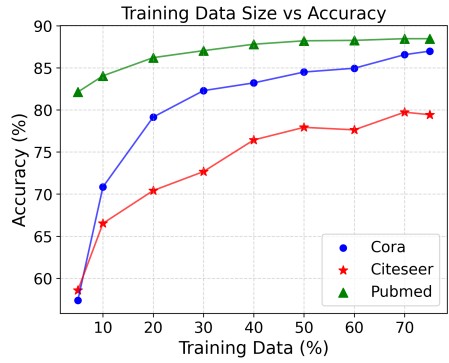

Figure 5: **Training set vs Accuracy**

## 5   Conclusion

In this work, we introduced SCNode, a novel method that integrates spatial and contextual information from graphs. Our results show that SCNode overcomes the predictive limitations of relying solely on either spatial or contextual features, achieving significant performance gains when both are informative. The model is computationally efficient and consistently outperforms or matches SOTA GNNs across diverse graph tasks, including small and large graphs as well as homophilic and heterophilic settings. Furthermore, the plug-and-play design of SCNode vectors highlights their flexibility, providing a powerful enhancement to existing GNN architectures. In future work, we plan to refine SCNode for deeper integration with GNNs and expand its application to temporal graphs by incorporating temporal dynamics alongside node attributes.

**Broader Impact Statement**

Our SCNode framework advances graph representation learning by addressing the fundamental limitations of GNNs, particularly their struggles with heterophilic graphs, underreaching, and oversquashing. By integrating spatial and contextual information through a novel landmark-based relative positioning approach, SCNode enhances the adaptability of GNNs across diverse datasets. This breakthrough enables more accu-

rate node classification and link prediction, improving applications in bioinformatics, social networks, and financial modeling. Additionally, SCNode embeddings serve as plug-and-play features, significantly boosting existing GNN models' performance, thereby bridging the gap between homophilic and heterophilic settings. This research contributes to the broader goal of more generalizable and interpretable graph learning models, expanding their impact across scientific and industrial domains.

**Acknowledgments**

This research was partially supported by the National Science Foundation under grants DMS-2220613 and DMS-2229417. The authors acknowledge the Texas Advanced Computing Center (TACC) at UT Austin for providing computational resources that have contributed to this paper.

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

# Appendix

## A   Homophily and SCNode

### A.1   Recent Homophily Metrics

Until recently, the prevailing homophily metrics were node homophily Pei et al. (2019) and edge homophily Abu-El-Haija et al. (2019); Zhu et al. (2020). Node homophily simply computes, for each node, the proportion of its neighbors that belong to the same class, and averages across all nodes, while edge homophily measures the proportion of edges connecting nodes of the same class compared to all edges in the network. In the past few years, to study heterophily phenomena in graph representation learning, several new homophily metrics were introduced, e.g., class homophily Lim et al. (2021b), generalized edge homophily Jin et al. (2022) and aggregation homophily Luan et al. (2022), adjusted homophily Platonov et al. (2023a) and label informativeness Platonov et al. (2023a). In Table 8, we give these metrics for our datasets. The details of these metrics can be found in Luan et al. (2023).

$$\mathcal{H}_{\text{node}}(\mathcal{G}) == \frac{1}{|\mathcal{V}|} \sum_{v \in \mathcal{V}} \mathcal{H}_{\text{node}}^v = \frac{1}{|\mathcal{V}|} \sum_{v \in \mathcal{V}} \frac{\left| \{u \mid u \in \mathcal{N}_v, Z_{u,:} = Z_{v,:}\} \right|}{d_v}$$

$$\mathcal{H}_{\text{edge}}(\mathcal{G}) = \frac{\left| \{e_{uv} \mid e_{uv} \in \mathcal{E}, Z_{u,:} = Z_{v,:}\} \right|}{|\mathcal{E}|}$$

$$\mathcal{H}_{\text{class}}(\mathcal{G}) = \frac{1}{C-1} \sum_{k=1}^{C} \left[ h_k - \frac{\left| \{v \mid Z_{v,k}=1\} \right|}{N} \right]_+ \text{ where } h_k = \frac{\sum_{v \in \mathcal{V}, Z_{v,k}=1} \left| \{u \mid u \in \mathcal{N}_v, Z_{u,:} = Z_{v,:}\} \right|}{\sum_{v \in \{v \mid Z_{v,k}=1\}} d_v}$$

$$\mathcal{H}_{\text{GE}}(\mathcal{G}) = \frac{\sum\limits_{(i,j) \in \mathcal{E}} \cos(x_i, x_j)}{|\mathcal{E}|}$$

$$\mathcal{H}_{\text{agg}}(\mathcal{G}) = \frac{1}{|\mathcal{V}|} \times \left| \left\{ v \mid \text{Mean}_u \left( \{ S(\hat{A}, Z)_{v,u}^{Z_{u,:}=Z_{v,:}} \} \right) \geq \text{Mean}_u \left( \{ S(\hat{A}, Z)_{v,u}^{Z_{u,:} \neq Z_{v,:}} \} \right) \right\} \right|$$

$$\mathcal{H}_{\text{adj}} = \frac{\mathcal{H}_{\text{edge}} - \sum_{c=1}^{C} \bar{p}_c^2}{1 - \sum_{k=1}^{C} \bar{p}_c^2}$$

$$\text{LI} = -\frac{\sum_{c_1,c_2} p_{c_1,c_2} \log \frac{p_{c_1,c_2}}{\bar{p}_{c_1} \bar{p}_{c_2}}}{\sum_c \bar{p}_c \log \bar{p}_c} = 2 - \frac{\sum_{c_1,c_2} p_{c_1,c_2} \log p_{c_1,c_2}}{\sum_c \bar{p}_c \log \bar{p}_c}$$

where $\mathcal{H}_{\text{node}}^v$ is the local homophily value for node $v$; $[a]_+ = \max(0, a)$; $h_k$ is the class-wise homophily metric Lim et al. (2021b); $\text{Mean}_u (\{\cdot\})$ takes the average over $u$ of a given multiset of values or variables and $S(\hat{A}, Z) = \hat{A} Z (\hat{A} Z)^\top$ is the post-aggregation node similarity matrix; $D_c = \sum_{v:z_v=c} d_v, \bar{p}_c = \frac{D_c}{2|\mathcal{E}|}$, $p_{c_1,c_2} = \sum_{(u,v) \in \mathcal{E}} \frac{1\{z_u=c_1, z_v=c_2\}}{2|\mathcal{E}|}, c, c_1, c_2 \in \{1, \dots, C\}$.

Table 8: Homophily metrics for our datasets.

| Metric | CORA | CITES. | PUBMED | TEXAS | CORNELL | WISC. | CHAM. |
|---|---|---|---|---|---|---|---|
| $\mathcal{H}_{SCN}$-spat-1 | 0.8129 | 0.6861 | 0.7766 | 0.1079 | 0.1844 | 0.2125 | 0.2549 |
| $\mathcal{H}_{SCN}$-context | 0.1702 | 0.1949 | 0.3245 | 0.2352 | 0.2409 | 0.2449 | 0.2564 |
| $\mathcal{H}_{\text{node}}$ | 0.8252 | 0.7175 | 0.7924 | 0.3855 | 0.1498 | 0.0968 | 0.2470 |
| $\mathcal{H}_{\text{edge}}$ | 0.8100 | 0.7362 | 0.8024 | 0.5669 | 0.4480 | 0.4106 | 0.2795 |
| $\mathcal{H}_{\text{class}}$ | 0.7657 | 0.6270 | 0.6641 | 0.0468 | 0.0941 | 0.0013 | 0.0620 |
| $\mathcal{H}_{\text{agg}}$ | 0.9904 | 0.9826 | 0.9432 | 0.8032 | 0.7768 | 0.6940 | 0.6100 |
| $\mathcal{H}_{\text{GE}}$ | 0.1700 | 0.1900 | 0.2700 | 0.3100 | 0.3400 | 0.3500 | 0.0152 |
| $\mathcal{H}_{\text{adj}}$ | 0.8178 | 0.7588 | 0.7431 | 0.1889 | 0.0826 | 0.0258 | 0.0663 |
| LI | 0.5904 | 0.4508 | 0.4093 | 0.0169 | 0.1311 | 0.1923 | 0.0480 |

Table 9: Homophily matrices for CORA dataset. **Every row** represents the corresponding homophily ratio of the row's class. In every row, the highest homophily ratio is marked **bold**.

| SCNode Spatial-1 Homophily Matrix | | | | | | | | SCNode Context-S Homophily Matrix | | | | | | | | SCNode Context-I Homophily Matrix | | | | | | |
|---|---|---|---|---|---|---|---|---|---|---|---|---|---|---|---|---|---|---|---|---|---|---|
| | $C_1$ | $C_2$ | $C_3$ | $C_4$ | $C_5$ | $C_6$ | $C_7$ | | $C_1$ | $C_2$ | $C_3$ | $C_4$ | $C_5$ | $C_6$ | $C_7$ | | $C_1$ | $C_2$ | $C_3$ | $C_4$ | $C_5$ | $C_6$ | $C_7$ |
| $C_1$ | **0.743** | 0.029 | 0.014 | 0.083 | 0.050 | 0.037 | 0.043 | $C_1$ | **0.206** | 0.143 | 0.126 | 0.111 | 0.124 | 0.143 | 0.147 | $C_1$ | **0.159** | 0.134 | 0.142 | 0.154 | 0.143 | 0.136 | 0.132 |
| $C_2$ | 0.040 | **0.769** | 0.062 | 0.080 | 0.020 | 0.028 | 0.002 | $C_2$ | 0.123 | **0.236** | 0.146 | 0.121 | 0.101 | 0.147 | 0.126 | $C_2$ | 0.142 | **0.159** | 0.146 | 0.154 | 0.139 | 0.138 | 0.122 |
| $C_3$ | 0.010 | 0.025 | **0.917** | 0.032 | 0.001 | 0.014 | 0.001 | $C_3$ | 0.109 | 0.171 | **0.253** | 0.107 | 0.088 | 0.136 | 0.137 | $C_3$ | 0.136 | 0.140 | **0.162** | 0.156 | 0.140 | 0.138 | 0.127 |
| $C_4$ | 0.055 | 0.020 | 0.016 | **0.839** | 0.051 | 0.015 | 0.004 | $C_4$ | 0.122 | 0.170 | 0.168 | **0.159** | 0.130 | 0.134 | 0.118 | $C_4$ | 0.145 | 0.136 | 0.146 | **0.168** | 0.151 | 0.133 | 0.122 |
| $C_5$ | 0.058 | 0.014 | 0.002 | 0.064 | **0.849** | 0.011 | 0.003 | $C_5$ | 0.130 | 0.154 | 0.118 | 0.110 | **0.240** | 0.137 | 0.111 | $C_5$ | 0.144 | 0.131 | 0.144 | 0.159 | **0.167** | 0.133 | 0.124 |
| $C_6$ | 0.058 | 0.017 | 0.030 | 0.051 | 0.018 | **0.786** | 0.040 | $C_6$ | 0.112 | 0.148 | 0.135 | 0.103 | 0.078 | **0.283** | 0.142 | $C_6$ | 0.142 | 0.132 | 0.145 | 0.149 | 0.145 | **0.158** | 0.129 |
| $C_7$ | 0.113 | 0.001 | 0.003 | 0.022 | 0.006 | 0.067 | **0.788** | $C_7$ | 0.150 | 0.146 | 0.123 | 0.093 | 0.097 | 0.154 | **0.236** | $C_7$ | 0.142 | 0.126 | 0.142 | 0.152 | 0.141 | 0.138 | **0.159** |

Table 10: Homophily matrices for the WISCONSIN dataset. **Every row** represents the corresponding homophily ratio of the row's class. In every row, the highest homophily ratio is marked **bold**.

| SCNode Spatial-1 Homophily Matrix | | | | | | SCNode Spatial-2 Homophily Matrix | | | | | | SCNode Domain Homophily Matrix | | | | |
|---|---|---|---|---|---|---|---|---|---|---|---|---|---|---|---|---|---|
| | $C_1$ | $C_2$ | $C_3$ | $C_4$ | $C_5$ | | $C_1$ | $C_2$ | $C_3$ | $C_4$ | $C_5$ | | $C_1$ | $C_2$ | $C_3$ | $C_4$ | $C_5$ |
| $C_1$ | **0.323** | 0.000 | 0.000 | 0.323 | 0.323 | $C_1$ | **0.444** | 0.278 | 0.111 | 0.111 | 0.056 | $C_1$ | **0.221** | 0.198 | 0.207 | 0.192 | 0.181 |
| $C_2$ | 0.001 | 0.219 | **0.595** | 0.185 | 0.001 | $C_2$ | 0.001 | **0.439** | 0.438 | 0.106 | 0.016 | $C_2$ | 0.148 | **0.247** | 0.223 | 0.201 | 0.181 |
| $C_3$ | 0.000 | **0.547** | 0.225 | 0.145 | 0.084 | $C_3$ | 0.011 | 0.351 | **0.481** | 0.111 | 0.046 | $C_3$ | 0.174 | 0.205 | **0.228** | 0.204 | 0.189 |
| $C_4$ | 0.044 | 0.268 | **0.468** | 0.071 | 0.150 | $C_4$ | 0.029 | 0.238 | **0.485** | 0.110 | 0.138 | $C_4$ | 0.145 | 0.203 | 0.217 | **0.237** | 0.199 |
| $C_5$ | 0.149 | 0.039 | **0.397** | 0.200 | 0.215 | $C_5$ | 0.149 | 0.084 | **0.374** | 0.136 | 0.258 | $C_5$ | 0.149 | 0.204 | 0.209 | 0.209 | **0.230** |

In Table 9 below, we give SCNode Homophily matrices which provides detailed insights on the class interactions in CORA dataset. In particular, Spatial-1 matrix interprets as for any class, the nodes likely to form a link with same class node in their one neighborhood. In the following Contextual-Selective and Contextual-Inclusive Homophily matrices, independent of graph distance, we see the positions of attribute vectors similarity to the chosen class landmarks.

## A.2 SCNode Homophily Matrices

We provide further homophily matrices for one homophilic and one heterophilic graphs. In Table 9, for CORA dataset, we observe in SCN Spatial-1, Context-S and Context-I Homophily matrices show the strong homophily behavior in both spatial and contextual aspects. In Spatial-1 matrix the entry $h_{ij}$ represents how likely the nodes in $\mathcal{C}_i$ to connect to nodes in $\mathcal{C}_j$ among their 1-neighborhood. The very high numbers in the diagonal shows that most classes likely to connect with their own class, as CORA's node homophily ratio (0.83) suggest. From the matrix, we have finer information that the class $\mathcal{C}_3$ is very highly homophilic. The domain matrices represents the average similarity/closeness to class landmarks. We observe that in both domain matrices, the node feature vectors likely to land close to their own class landmark, and SCNode vectors captures this crucial information.

In Table 10, for WISCONSIN dataset, spatial-1 and Spatial-2 matrices show the irregular behavior as the WISCONSIN node homophily ratio (0.09) suggest. The nodes are unlikely to connect their adjacent nodes in the same class. Even in their two neighborhoods, they don't have many of their fellow classmate for $\mathcal{C}_4$ and $\mathcal{C}_5$. However, while the other three classes are not connecting their own classmate, they have several common neighbors. Finally, the domain homophily matrix again shows that while they are not very close in the graph, they all share common interests, as every node's feature vector lands close to their classmates'.

While different homophily ratios are summarizing crucial information about node tendencies, we observe that our matrices are giving much finer and easily intepretable information no class behaviors.

## A.3 Proof of the Theorem and New Homophily Metrics

In Section 3.4, we have defined homophily and discussed the relation between SCNode vectors and the homophily notion. Here we give details how to generalize this idea to give finer homophily notions for graphs. First, we give the proof of Theorem 3.7.

**Theorem A.1.** *For an undirected graph $\mathcal{G} = (\mathcal{V}, \mathbb{E})$, let $\vec{\alpha}_1(v)$ be the spatial vector defined in Appendix A. Let $\widehat{\alpha}_1(v)$ be the vector where the entry corresponding to class of $v$ is set to $0$. Then,*

$$1 - \varphi(\mathcal{G}) = \frac{1}{|\mathcal{V}|} \sum_{v \in \mathcal{V}} \frac{\|\widehat{\alpha}_1(v)\|_1}{\|\vec{\alpha}_1(v)\|_1}$$

*Proof.* For an undirected graph $\mathcal{G} = (\mathcal{V}, \mathbb{E})$ with node class assignment function $\mathcal{C} : \mathcal{V} \to \{1, 2, \ldots, N\}$, $\vec{\alpha}_1(v) = [a_1(v)\ a_2(v)\ \ldots\ a_N(v)]$ where $a_i(v) = \#\{u \in \mathcal{N}_1(v) \mid \mathcal{C}(u) = i\}$. Let $\mathcal{C}(v) = j_v$. Then, by setting $a_{j_v}(v) = 0$, we get a new vector $\widehat{\alpha}_1(v) = [a_1(v) \ldots\ a_{j_v-1}(v)\ 0\ a_{j_v+1}(v)\ \ldots]$

$L^1$-norm of $\vec{\alpha}_1(v)$ is $\|\vec{\alpha}_1(v)\|_1 = \sum_{i=1}^{N} a_i(v)$. Similarly, $\|\widehat{\alpha}_1(v)\|_1 = \sum_{i \neq j_v} a_i(v)$. Hence, we have $\|\vec{\alpha}_1(v)\|_1 - \|\widehat{\alpha}_1(v)\|_1 = a_{j_v}(v)$. Notice that by definition, $a_{j_v}(v) = \eta(v)$, the number of neighbors in the same class with $v$. Similarly, $\|\vec{\alpha}_1(v)\|_1 = deg(v)$. Hence,

$$1 - \frac{\|\widehat{\alpha}_1(v)\|_1}{\|\vec{\alpha}_1(v)\|_1} = \frac{\|\vec{\alpha}_1(v)\|_1 - \|\widehat{\alpha}_1(v)\|_1}{\|\vec{\alpha}_1(v)\|_1} = \frac{\eta(v)}{deg(v)} \tag{1}$$

As $\varphi(\mathcal{G}) = \frac{1}{|\mathcal{V}|} \sum_{v \in \mathcal{V}} \frac{\eta(v)}{deg(v)}$, we have $1 - \varphi(\mathcal{G})$ as

$1 - \frac{1}{|\mathcal{V}|} \sum_{v \in \mathcal{V}} \frac{\eta(v)}{deg(v)} = \frac{1}{|\mathcal{V}|} \sum_{v \in \mathcal{V}} (1 - \frac{\eta(v)}{deg(v)}) = \frac{1}{|\mathcal{V}|} \sum_{v \in \mathcal{V}} \frac{\|\widehat{\alpha}_1(v)\|_1}{\|\vec{\alpha}_1(v)\|_1}$

where the last equality follows by Equation (1). The proof follows. $\qquad\square$

This perspective inspires different ways to generalize the homophily concept by using SCNode vectors. Notice that in SCNode matrices above, we employed a classwise grouping to measure class interactions. If we do not use any grouping for the nodes, we get natural generalizations of existing homophily ratios. First, we define higher homophily by using the ratio of the number of nodes in the 2-neighborhood $\mathcal{N}_2(v)$ with the same class to $|\mathcal{N}_2(v)|$.

**Definition A.2** (Higher Homophily)**.** Given $\mathcal{G} = (\mathcal{V}, \mathbb{E})$ with $\mathcal{C} : \mathcal{V} \to \{1, 2, \ldots, N\}$ representing node classes. Let $\eta_2(v)$ be the number of nodes in $\mathcal{N}_2(v)$ in the same class with $v$. Then, homophily ratio of $\mathcal{G}$ is defined as

$$\varphi_2(\mathcal{G}) = \frac{1}{|\mathcal{V}|} \sum_{v \in \mathcal{V}} \frac{\eta_2(v)}{|\mathcal{N}_2(v)|}$$

By applying similar ideas to the proof of Theorem 3.7, we obtain the following result.

**Theorem A.3.** *For an undirected graph $\mathcal{G} = (\mathcal{V}, \mathbb{E})$, let $\vec{\alpha}_2(v)$ be the spatial vector defined above. Let $\widehat{\alpha}_2(v)$ be the vector where the entry corresponding to the class of $v$ is set to $0$. Then,*

$$1 - \varphi_2(\mathcal{G}) = \frac{1}{|\mathcal{V}|} \sum_{v \in \mathcal{V}} \frac{\|\widehat{\alpha}_2(v)\|_1}{\|\vec{\alpha}_2(v)\|_1}$$

While the above notions represent structural homophily, we introduce another homophily by comparing the contextual vectors of neighboring nodes with the central node's.

**Definition A.4** (Contextual Homophily)**.** Given $\mathcal{G} = (\mathcal{V}, \mathbb{E})$ with $\mathcal{C} : \mathcal{V} \to \{1, 2, \ldots, N\}$ representing node classes. Let $\vec{\beta}(v)$ be the contextual vector as defined in Section 3.2. Let $\widehat{\beta}(v)$ be the vector where the entry corresponding to the class of $v$ is set to $0$. Then, define the contextual homophily of $\mathcal{G}$ as

$$\varphi_{\mathcal{D}}(\mathcal{G}) = 1 - \frac{1}{|\mathcal{V}|} \sum_{v \in \mathcal{V}} \frac{\|\widehat{\beta}(v)\|_1}{\|\vec{\beta}(v)\|_1}$$

Similar homophily metrics, as introduced in Luan et al. (2023); Lim et al. (2021b); Zhu et al. (2020); Jin et al. (2022), were employed to investigate the concept of heterophily in graph representation learning. In

particular, our SCNode features inherently encompass this information, enabling the ML classifier to adapt accordingly. This explains the outstanding performance of SCNode in both homophilic and heterophilic settings for node classification and link prediction.

*Remark* A.5 (SCNode and Underreaching). Underreaching arises because a depth-$K$ message-passing GNN is $K$-local: its output at node $u$ depends only on the $K$-hop ego-network and features within it, so nodes that are $K$-hop isomorphic remain indistinguishable. SCNode breaks this locality at the input level before message passing.

From a *spectral perspective*, spatial coordinates $\alpha_k(u)$ are class-distribution statistics over $K$-hop neighborhoods and can be expressed as $\alpha_k = P_k(A)Y$, where $P_k$ is a degree-$K$ polynomial in the normalized adjacency $A$ and $Y$ encodes class indicators on labeled nodes. Thus, $\alpha_k$ lies within the span of polynomial filters in $A$. In contrast, contextual coordinates $\beta(u) = [d(X_u, \xi_1), \ldots, d(X_u, \xi_N)]$ are distances to class landmarks $\{\xi_j\}$ in attribute space and are independent of $A$. Concatenating $[\alpha_k(u) \, \| \, \beta(u)]$ therefore injects a nonlocal, zero-hop channel that cannot be reproduced by any finite-degree polynomial in $A$, strictly enlarging the representable function class.

From an *expressivity perspective*, two nodes that are $K$-hop isomorphic receive identical inputs under any $K$-local MPGNN. With contextual embeddings $\beta(u)$, such nodes can still be distinguished if they occupy different positions relative to class landmarks in attribute space, allowing even a shallow classifier on $[\alpha_k \, \| \, \beta]$ to separate them.

Empirically, Table 7 supports this analysis: contextual-only embeddings succeed in heterophilic graphs where local neighborhoods are misleading, while spatial-only embeddings are strongest on homophilic graphs. Their combination consistently achieves the best results, aligning with the theoretical claim that $\beta$ supplies global class information inaccessible to local filters, while $\alpha$ exploits reliable neighborhood structure.

# B  Generalizations of SCNode

## B.1  Inductive and Transductive Settings

**Inductive Setting.** In the inductive learning framework, a model is trained by using the training data $\mathcal{D}_{train}$ while the test data, $\mathcal{D}_{test}$ is completely hidden during the training time. This means that no information about test nodes (e.g., edges between training and test nodes) is provided during the training stage. The learning procedure aims to minimize a suitable loss function to capture the statistical distribution of the training data. Once an inductive model has undergone training, it can be utilized to make predictions on new (unseen) data, thereby determining the labels for unlabeled nodes.

**Transductive Setting.** In the transductive learning framework, which closely aligns with semi-supervised learning, both the training data $\mathcal{D}_{train}$ and the test data $\mathcal{D}_{test}$ can be simultaneously leveraged to capitalize on their interconnectedness. This interrelationship can be employed either during the training phase, the prediction phase, or both. Specifically, in the training stage, the information about $v_{test}$ and its position in the graph is known, while its label $y_{test}$ remains concealed. Consequently, the model is trained with explicit awareness of the nodes it will be evaluated on after the training process. This can serve as a valuable asset for the model, enabling it to establish a sound decision function by exploiting the characteristics observed in $v_{test}$.

To clarify the differences between the inductive and transductive settings in graph-based learning, consider a given graph $\mathcal{G}$, with datasets $\mathcal{D}_{\text{train}}$ and $\mathcal{D}_{\text{test}}$. In the inductive setting, all test nodes and their connected edges are removed to create the training subgraph $\mathcal{G}_{\text{train}}$. The model is trained exclusively on $\mathcal{G}_{\text{train}}$ and only gains access to the complete graph $\mathcal{G}$ during the testing stage. Conversely, in the transductive learning approach, the model has access to the entire graph $\mathcal{G}$ during training; however, the labels of the test nodes remain hidden throughout this process. It is noteworthy that any dataset configured for transductive learning can be adapted to the inductive setting by excluding test nodes and their connecting edges during the training phase. However, converting from the inductive to the transductive setting is not generally feasible. For further details, see the references Ciano et al. (2021); Arnold et al. (2007).

## B.2 SCNode for Transductive Setting and Iterated Predictions

So far, we outlined our SCNode vectors for simplicity in inductive setting. To adapt to the transductive setting, we make adjustments without altering contextual vectors. In transductive setting, test node labels are hidden during training, but connection information to training nodes is available. We introduce a new "unknown" class $\mathcal{C}_u$ for test nodes with unknown labels, considering them as neighbors to training nodes. Each node $u \in \mathcal{V}$ is represented by a $N+1$ dimensional vector $\vec{\alpha}_1^0(u) = [a_{10}^0 \quad a_{11}^0 \quad a_{12}^0 \quad \ldots \quad a_{1N}^0]$, where $a_{10}^0$ is the count of neighboring test nodes (unknown labels) of node $u$, and $a_{1j}^0$ is the count of neighboring training nodes in class $\mathcal{C}_j$ for $1 \leq j \leq N$. Similar representations are defined for $\vec{\alpha}_{1i}(u)$, $\vec{\alpha}_{1o}(u)$, and $\vec{\alpha}_2(u)$. The superscript 0 indicates no iterations have occurred yet.

**Iterated Predictions:** Recall that the ultimate goal in the node classification problem is to predict the labels of new (test) nodes. After we obtain all SCNode vectors for training and test nodes above, we let our ML model make a prediction for each test node $v \in \mathcal{V}_{test}$. Let $\mathcal{P}_0 : \mathcal{V}_{test} \to \{\mathcal{C}_1, \ldots, \mathcal{C}_N\}$ be our predictions. Hence, we have a label for each node in our graph $\mathcal{G}$.

*While the original spatial vector $\vec{\alpha}_1^0(u)$ cannot use any class information for the test nodes, in the next step, we remedy this by using our class predictions for test nodes.* In particular, by using the predictions $\mathcal{P}_0$, we define a new (improved) spatial vector $\vec{\alpha}_1^1(u) = [a_{11}^1 \quad a_{12}^1 \quad a_{13}^1 \quad \ldots \quad a_{1N}^1]$ where we use predictions $\mathcal{P}_0$ for neighboring test nodes. Notice that this is no longer a $(N+1)$-dimensional vector as there is no unknown class anymore. Similarly, we update all spatial vectors, train our ML model with these new node labels, and make a new prediction for test nodes. Then, we get new label predictions $\mathcal{P}_1 : \mathcal{V}_{test} \to \{\mathcal{C}_1, \ldots, \mathcal{C}_N\}$. By using predictions $\mathcal{P}_1$, we define the next iteration $\vec{\alpha}_1^2(u) = [a_{11}^2 \quad a_{12}^2 \quad \ldots \quad a_{1N}^2]$ and train our model with these updated vectors. Again, we get new predictions $\mathcal{P}_2 : \mathcal{V}_{test} \to \{\mathcal{C}_1, \ldots, \mathcal{C}_N\}$. In our experiments, we observe that 1 or 2 iterations ($\mathcal{P}_1$ or $\mathcal{P}_2$) improve the performance significantly, but further iterations do not, in the transductive setting.

## B.3 Spatial Embeddings on Directed, Weighted Graphs

**Directed Graphs.** When $\mathcal{G}$ is directed, to obtain finer information about node neighborhoods, we produce two different embeddings of size $N$, $\vec{\alpha}_{ki}(u)$ and $\vec{\alpha}_{ko}(u)$ for the k-hop neighborhood:

$$\vec{\alpha}_{ki}(u) = [a_{k1}^i \quad a_{k2}^i \quad \ldots \quad a_{kN}^i] \quad \vec{\alpha}_{ko}(u) = [a_{k1}^o \quad a_{k2}^o \quad \ldots \quad a_{kN}^o]$$

where $a_{kj}^i$ is the count of k-hop neighbors incoming to $u$ belonging to class $\mathcal{C}_j$ while $a_{kj}^o$ is the count of neighbors outgoing from $u$ belonging to class $\mathcal{C}_j$ (See Figure 2(a)).

**Weighted Graphs:** In weighted graphs, the counts incorporate edge weight information. Specifically, we define the weighted feature vector $\vec{\alpha}_k^w(u)$ for node $u$ as: $\vec{\alpha}_k^w(u) = [a_{k1}^w, a_{k2}^w, a_{k3}^w, \ldots, a_{kN}^w]$ where $a_{kj}^w$ is the sum of the weights of the edges connecting $u$ to the $k$-hop neighbors belonging to class $\mathcal{C}_j$. If the weight of an edge is inversely proportional to the similarity between nodes, one can use the sum of the reciprocals of the weights instead: $a_{kj}^w = \sum_{v \in \mathcal{N}_k(u) \cap \mathcal{C}_j} \frac{1}{\text{weight}(u,v)}$. This approach allows for adjustments in how edge weights influence the calculation of proximity and class association. Additionally, in the context of directed graphs within a weighted graph setting, different vectors can be defined for incoming and outgoing neighborhood connections, such as $\vec{\alpha}_{ki}^w(u)$ for incoming edges and $\vec{\alpha}_{ko}^w(u)$ for outgoing edges, providing a more detailed representation of node relationships based on directionality and weight.

Table 11: The dimension of the SCNode embedding used for each dataset in our model

| | CORA | CITESEER | PUBMED | OGBN-ARXIV | OGBN-MAG | TEXAS | CORNELL | WISCONSIN | CHAMELEON |
|---|---|---|---|---|---|---|---|---|---|
| Dimension | 42 | 36 | 112 | 120 | 1,524 | 30 | 30 | 30 | 20 |

## B.4 Further Ablation Studies

In this part, we present further ablation studies to evaluate the sensitivity of SCNode to neighborhood size in spatial embeddings and to similarity metrics in contextual embeddings. Recall that the two embeddings

serve distinct roles: (1) **Spatial embeddings** are derived from neighborhood label distributions up to a fixed hop size and do not involve landmarks; (2) **Contextual embeddings** are constructed using class landmarks in attribute space, with similarity metrics selected according to feature type (e.g., Jaccard for sparse binary features such as Cora and Citeseer, Euclidean for real-valued features such as PubMed).

**Spatial Embeddings: Effect of Neighborhood Size.**   We compared spatial embeddings derived from 1-hop and 2-hop neighborhoods. Results show that 2-hop neighborhoods yield modest accuracy improvements across datasets, though at a higher computational cost. For example, on PubMed the accuracy increases from $77.59 \pm 0.61$ with 1-hop to $82.28 \pm 0.09$ with 2-hop, while runtime grows from 63s to 232s. Based on this trade-off, we adopt 2-hop neighborhoods in the main experiments as a balanced choice. The detailed results are reported in Table 12.

**Contextual Embeddings: Effect of Similarity Metrics.**   We further evaluated contextual embeddings using Jaccard, cosine, and Euclidean metrics. On **binary datasets** (Cora, Citeseer), Jaccard provides the best performance, while cosine achieves similar results but at higher computational cost. Euclidean performs substantially worse in this setting, as expected for sparse binary vectors. On **real-valued datasets** (PubMed), both Euclidean and cosine produce strong results. These findings confirm that the effectiveness of contextual embeddings depends on aligning the similarity metric with the feature type. Full results are provided in Table 13.

These ablations demonstrate that SCNode is robust to landmark quantity and neighborhood size, and that contextual similarity metrics should be selected according to feature type to achieve the best balance between accuracy and efficiency.

Table 12: **Effect of Neighborhood Size.** Performances of spatial-only vectors used by different k-hop neighborhoods.

| Ngbd Size | Accuracy | | | Time | | |
|---|---|---|---|---|---|---|
| | **Cora** | **Citeseer** | **PubMed** | **Cora** | **Citeseer** | **PubMed** |
| **1-hop** | $83.96 \pm 1.35$ | $71.28 \pm 1.84$ | $77.59 \pm 0.61$ | 1 s | 1 s | 63 s |
| **2-hop** | $84.61 \pm 1.20$ | $71.82 \pm 1.95$ | $82.28 \pm 0.09$ | 3 s | 3 s | 232 s |

Table 13:   **Effect of Metric Choice.** Performances of contextual-only vectors obtained by different similarity metrics.

| Metric | Accuracy | | | Time | | |
|---|---|---|---|---|---|---|
| | **Cora** | **Citeseer** | **PubMed** | **Cora** | **Citeseer** | **PubMed** |
| Jaccard | $73.98 \pm 2.56$ | $67.42 \pm 0.85$ | n/a | 2 s | 7 s | n/a |
| Cosine | $73.50 \pm 2.48$ | $67.09 \pm 1.80$ | $85.62 \pm 0.41$ | 5 s | 15 s | 17 s |
| Euclidean | $62.60 \pm 2.65$ | $60.89 \pm 1.75$ | $85.92 \pm 0.52$ | 2 s | 6 s | 17 s |

# C   Details of SCNode Embeddings

In all datasets, we basically used the same method to obtain our vectors, however, when the graph type (directed, undirected), or node attribute vector format varies, our methodology naturally adapts the corresponding setting as detailed below. Note that details of OGBN datasets can be found at Hu et al. (2020) and at Stanford's Open Graph Benchmark site [1].

**CORA:** The CORA dataset is a directed graph of scientific publications classified into one of the 7 classes. Each node $u$ represents a paper and comes with a binary (0/1) vector $X_u$ of length 1433 indicating the presence/absence of the corresponding word in the paper from a dictionary of 1433 unique words. The

---

[1]`https://ogb.stanford.edu/docs/nodeprop/`

task is to predict the subject area of the paper, e.g. Neural Networks, Theory, Case-Based. The directed graph approach is used to extract the attribute vector, $\gamma^0(u)$, from the CORA dataset. Recall that this is transductive setting, and the first row is an 8 dimensional spatial vector $\alpha_{1i}^0(u)$ (Section 3.1) where the first 7 entries represent the count of citing (incoming) papers from the corresponding class and the $8^{th}$ entry is the count of unknown citing paper ($v_i \in D_{test}$) (Appendix B.2). The second row $\alpha_{1o}^0(u)$ is defined similarly using the count of cited (outgoing) papers of the corresponding class. The third and fourth row of the spatial feature vector is obtained similarly using the second neighborhood information of each node for citing and cited paper respectively. For the first iteration $\gamma^1(u)$, the same setup is followed ignoring the $8^{th}$ entry of each row because there is no unknown class now.

For the contextual vector $\vec{\beta}_1(u)$, we follow the landmark approach described in Section 3.2. We define the first landmark vector $\xi_j^1$ which is a binary word vector of length 1433 such that the entry is 1 if the corresponding word is present in any of binary vectors $\mathcal{X}_u$ belonging to class $\mathcal{C}_j$, and the entry is 0 otherwise. Then, each entry $b_{1j}$ of $\vec{\beta}_1(u)$ is the count of common words between $\mathcal{X}_u$ and $\xi_j^1$ for each class, which produces a 7 dimensional contextual vector. Similarly, for the contextual vector $\vec{\beta}_2(u)$, we use a more selective landmark vector $\xi_j^2$ is defined as a binary word vector of length 1433 indicating the presence/absence of the corresponding word in at least 10% nodes in the class $\mathcal{C}_j$. Hence, the initial vector $\gamma^0(u)$ is 46 dimensional (32 spatial, 14 contextual), in the next iterations, $\gamma^1(u), \gamma^2(u)$ are both 42-dimensional (28 spatial, 14 contextual).

**CITESEER:** The CITESEER is also a directed graph of scientific publication classified into one of the 6 classes. Each node represents a paper and comes with a binary vector like CORA from a dictionary of 3703 unique words. Here, the aim of the node classification task is to make a prediction about the subject area of the paper. Since the graph properties/structure and node representing word vector is similar to the CORA dataset, the same feature-extracting techniques for both spatial and contextual vectors is followed here. Hence, the initial vector $\gamma^0(u)$ is 40 dimensional (28 spatial, 12 contextual), in the next iterations, $\gamma^1(u), \gamma^2(u)$ are both 36-dimensional (24 spatial, 12 contextual).

**PUBMED:** The PUBMED dataset is a directed graph of 19717 scientific publications from the PubMed database pertaining to diabetes classified into one of three classes. Each node represents a publication and is described by a TF/IDF weighted word vector from a dictionary which consists of 500 unique words. Since the graph structure is quite similar to CORA and CITESEER, a similar method is followed to extract the spatial features. So the initial spatial vector is 16 dimensional and it is 12 dimensional for the second iteration. For the contextual vector $\vec{\beta}(u)$, Principal component analysis (PCA) is used to reduce the dimension of the given weighted word vector from 500 to 100. Hence, the initial vector, $\gamma^0(u)$ is 16 dimensional (spatial only) and the vector is 112 dimensional (12 spatial, 100 contextual) in the next iteration.

**OGBN-ARXIV:** The OGBN-ARXIV dataset is a directed graph, representing the citation network between all Computer Science (CS) arXiv papers indexed by MAG Wang et al. (2020). Each node is an arXiv paper and each directed edge indicates that one paper cites another one. Each paper comes with a 128-dimensional vector obtained by averaging the embeddings of words in its title and abstract. The embeddings of individual words are computed by running the skip-gram model Mikolov et al. (2013) over the MAG corpus. We also provide the mapping from MAG paper IDs into the raw texts of titles and abstracts here. In addition, all papers are also associated with the year that the corresponding paper was published. The task is to predict the 40 subject areas of arXiv CS papers, e.g., cs.AI, cs.LG, and cs.OS.

The vector $\gamma(u)$ for OGBN-ARXIV is obtained by using our directed graph approach (Section 3.1). The first row of spatial vector $\alpha_{1i}(u)$ is 40-dimensional, and each entry is the count of citing (incoming) papers from the corresponding class. The second row is $\alpha_{1o}(u)$ and is defined similarly, where each entry is the count of cited (outgoing) papers from the corresponding class. For contextual vector $\beta(u)$, follow the landmark approach described in Section 3.2, employing just one landmark. Since the vectors are weighted vector, we use Euclidean distance to determine the distance between the landmark and a given node. Hence, for each node $\gamma(u)$ has 80-dimensional spatial, and 40-dimensional contextual vector, which totals 120-dimensional SCNode vector.

**OGBN-MAG:** The OGBN-MAG dataset is a heterogeneous network composed of a subset of the Microsoft Academic Graph (MAG) Wang et al. (2020). It contains four types of entities—papers (736,389 nodes), authors (1,134,649 nodes), institutions (8,740 nodes), and fields of study (59,965 nodes)—as well as four types of directed relations connecting two types of entities—an author is "affiliated with" an institution, an author "writes" a paper, a paper "cites" a paper, and a paper "has a topic of" a field of study. Similar to OGBN-ARXIV, each paper is associated with a 128-dimensional word2vec vector, and all the other types of entities are not associated with input node features. Given the heterogeneous OGBN-MAG data, the task is to predict one of 349 venues (conference or journal) of each paper.

The vector $\gamma(u)$ for OGBN-MAG is a bit different than OGBN-ARXIV, as OGBN-MAG is a heterogeneous network. We first collapse the network to a homogeneous network for papers. Similar to OGBN-ARXIV, we obtain 349-dimensional spatial vectors, i.e., $\alpha_{1i}(u)$ (citing papers), and $\alpha_{1o}(u)$ (cited papers). As another spatial vector from a different level of the heterogeneous network, we use author information as follows. Each author has a natural 349-dimensional vector where each entry is the number of papers the author published in the corresponding venue. For each paper, we consider the author with the most publications and assign their attribute vector to the paper's attribute vector. We call it author vector $\alpha_{author}(u)$. We construct a similar set of vectors for field of study - another type of node information. Each paper belongs to 1 or more fields of studies (or topics), and for each unique topic, we construct a attribute vector $T = \{t_1, t_2, ..., t_{num\_cls}\}$ such that $t_i$ denotes the number of papers assigned to venue $i$ for the given topic. We then aggregate these topic attribute vectors for each paper as follows: for a given paper with assigned topics $topic_1...topic_m$, let $\alpha_{topic}(u) = \sum_{i=1}^{m} T_i$ and append this final aggregate vector to the paper's attribute vector. For contextual vector $\beta(u)$, we directly use a 128-dimensional vector for each node as it is. Hence, $\gamma(u)$ is concatenation of spatial vectors $\alpha_{1i}(u)$, $\alpha_{1o}(u)$, $\alpha_{author}(u)$, $\alpha_{topic}(u)$, and $\beta(u)$ which totals $4 \cdot 349 + 128 = 1524$ dimensional vector.

**WebKB (TEXAS, CORNELL and WISCONSIN):** Carnegie Mellon University collected the WebKB dataset from computer science departments of various universities. Three subsets of this dataset are TEXAS, CORNELL and WISCONSIN. The dataset contains links between web pages, indicating relationships like "is located in" or "is a student of" forming a directed graph structure. Node features are represented as bag-of-words, creating a binary vector for each web page. The classification task involves categorizing nodes into five types: student, project, course, staff, and faculty. Similar to the CORA dataset, these datasets share a directed graph structure and binary vector representation for node features, leading to the use of comparable feature extraction methods for spatial and contextual vectors. Therefore, the initial attribute vector $\gamma^0(u)$ is 34 dimensional, comprising 24 spatial and 10 contextual dimensions. In subsequent iterations, it is 30-dimensional, with 20 spatial and 10 contextual dimensions each.

**Wikipedia Network (CHAMELEON):** The dataset depict page-page networks focused on specific topics such as chameleons. In these networks, nodes represent articles, and edges denote mutual links between them. Node features are derived from several informative nouns found in the corresponding Wikipedia pages. If a feature is present in the feature list, it signifies the occurrence of an informative noun in the text of the Wikipedia article. The objective is to classify the nodes into five categories based on the average monthly traffic of the respective web pages. In the context of the Wikipedia network, each link between articles does not imply a one-way relationship; instead, it signifies a mutual connection between the two articles, making it an undirected graph. Therefore, undirected feature extraction approaches are employed for $\gamma^0(u)$.

Regarding spatial features, the first row consists of a 6-dimensional vector. The first five entries represent the count of five classes, while the sixth entry represents the count of the unknown class in the 1-hop neighborhood. In the second row, the same procedure is applied for the 2-hop neighborhood. Subsequent iterations follow a similar process, but the $6^{th}$ entries are ignored because there is no unknown class at this point in the analysis. For contextual features in the Wikipedia network, a similar approach as used for the CORA dataset is employed due to their analogous nature. This approach results in a 10-dimensional vector. Consequently, the initial vector is 22-dimensional, and subsequent iterations reduce it to 20-dimensional.

