# OpenReview forum: "SCNode: Spatial and Contextual Coordinates for Graph Representation Learning"
_TMLR — Accepted by TMLR_

### Review · Reviewer_nKnE · 2025-07-22

**Summary Of Contributions:**

The paper introduces SCNode, a framework that enhances node representations in graphs by integrating both spatial coordinates (from local neighborhoods) and contextual coordinates (based on distances to class-specific landmarks). This dual representation allows SCNode to perform robustly in both homophilic and heterophilic graphs, overcoming limitations of traditional GNNs such as oversquashing, oversmoothing, and underreaching. SCNode achieves state-of-the-art results in node classification and link prediction across diverse benchmark datasets.

**Audience:**

Yes

**Broader Impact Concerns:**

I do not have any concern.

**Claims And Evidence:**

Yes

**Requested Changes:**

- Could the authors discuss the similarity of the proposed method with feature augmentation techniques and kernel-based methods?

**Strengths And Weaknesses:**

**Strengths**
- SCNode consistently outperforms or matches the best models in both types of graph structures, where many GNNs struggle.
- The proposed method is evaluated in several datasets and settings.


**Weakness**
- While SCNode introduces a novel and effective framework for generating spatial and contextual node embeddings, the methodology bears a strong conceptual resemblance to traditional graph kernel approaches, particularly those that rely on structural similarity and relative positioning—such as Weisfeiler-Lehman kernels, shortest-path kernels, or graphlet kernels. Like SCNode, these methods transform node neighborhoods and relational structures into high-dimensional representations to facilitate downstream tasks.
However, the paper does not include any discussion or empirical comparison with kernel-based methods, nor are such approaches mentioned in the related work. This omission creates a gap in the theoretical framing and historical positioning of the work.

---

> ### Author Response · Authors · 2025-08-20
> **Response to Reviewer nKnE**
>
> ### **W1. Comparison with Kernel-base Method**
> >While SCNode introduces a novel and effective framework for generating spatial and contextual node embeddings, the methodology bears a strong conceptual resemblance to traditional graph kernel approaches, particularly those that rely on structural similarity and relative positioning—such as Weisfeiler-Lehman kernels, shortest-path kernels, or graphlet kernels. Like SCNode, these methods transform node neighborhoods and relational structures into high-dimensional representations to facilitate downstream tasks. However, the paper does not include any discussion or empirical comparison with kernel-based methods, nor are such approaches mentioned in the related work. This omission creates a gap in the theoretical framing and historical positioning of the work.
>
> Thank you for raising this important point. We agree that SCNode is conceptually related to WL-style features and kernel families (shortest-path, graphlet) that build fixed representations of neighborhood structure. They are indeed similar in the sense that SCNode, like kernels and feature augmentation, constructs fixed descriptors from local structure and attributes to aid downstream classifiers. However, SCNode is fundamentally different from kernel methods as follows:
>
> **Key differences**.
>
> **Class-aware by design**: Spatial uses class distributions over k-hop neighborhoods and contextual uses distances to class landmarks. Classical kernels are typically class-agnostic.
>
>
> **“Zero-hop” global context**: Contextual coordinates inject nonlocal information not recoverable by finite-depth WL or polynomial filters, addressing underreaching (Remark A.5).
>
>
> **Inductive and scalable**: SCNode yields explicit per-node vectors, avoiding n×n Gram matrices and supporting new nodes without retraining.
>
>
> **Plug-in with GNNs**: SCNode concatenates with common backbones and yields consistent gains, which kernel similarities do not provide out of the box (Table 4).
>
>
> **Attribute-aware**: Contextual coordinates operate in attribute space with metrics matched to feature type, which is crucial on heterophilic graphs (Table 7).
>
>
> We have added a discussion in the related work section highlighting the differences between graph-kernel methods and SCNode. Thank you for this helpful suggestion.

---

### Review · Reviewer_y9rn · 2025-08-05

**Summary Of Contributions:**

In this work, the authors identify key limitations of message-passing GNNs—namely oversquashing, oversmoothing, underreaching, and reliance on homophily—and propose SCNode, a hybrid embedding framework that augments each node with (1) spatial coordinates (counts of labeled neighbors up to two hops) and (2) contextual coordinates (distances to learned class landmarks in feature space). They introduce class-aware homophily matrices to analyze class interactions, demonstrate that SCNode yields state-of-the-art (SOTA) results on both homophilic and heterophilic benchmarks, and show that SCNode embeddings drop in as plug-and-play features to improve existing GNNs—with negligible extra training cost compared to multi-day baselines.

**Audience:**

Yes

**Claims And Evidence:**

Yes

**Requested Changes:**

1. Ablation on Landmark Quantity & Metrics: Evaluate SCNode with 1–k landmarks and additional similarity measures (e.g., cosine, learned prototypes), to gauge sensitivity.

2. Heterogeneous Graph Treatment: Provide more principled handling (e.g., relational landmarks) rather than flattening authors/topics into vote vectors for OGBN-MAG.

3. Runtime Breakdown: Detail memory and parallelization overhead for spatial/contextual computations, especially on graphs with large amounts of nodes.

4. Benchmark Against Newer Models: Include comparisons to the latest heterophily-focused GNNs (e.g., H2GCN 2024, GraphAny 2024) to ensure novelty.

5. Theoretical Insights: Expand on why the concatenation of spatial/contextual coordinates provably overcomes underreaching, possibly via spectral or expressivity analysis.

**Strengths And Weaknesses:**

**Strengths**

1. Clear Addressing of GNN Pathologies: By explicitly modeling spatial and contextual cues, SCNode tackles oversmoothing/undersquashing and heterophily in a unified way.

2. Strong Empirical Gains: Outperforms prior methods on homophilic benchmarks and heterophilic ones, with consistent improvements when plugged into standard GNNs.


**Weaknesses**

1. Dependence on Label Quality: Contextual embeddings require representative class centroids; poor or imbalanced labels could degrade performance.

2. Limited Ablation on Landmark Choices: Only two landmark types (Euclidean/Jaccard) are explored; impact of number and selection method of landmarks is not fully studied.

3. Scalability to Massive Heterogeneous Graphs: While OGBN-MAG is handled, the approach to collapse heterogeneous relations into paper-only views may lose information and the dimension may be prohibitive in larger knowledge graphs.

---

> ### Author Response · Authors · 2025-08-20
> **Response to Reviewer y9rn-part 1**
>
> *We sincerely thank the reviewer for their thoughtful reading and valuable feedback on our submission. We have carefully considered each comment and provide detailed responses, along with clarifications and additional experiments, to highlight the robustness and novelty of our approach. We hope our responses address all concerns and improve the clarity of our work.*
>
> ### **W1. Dependence on Label Quality**
> >Dependence on Label Quality: Contextual embeddings require representative class centroids; poor or imbalanced labels could degrade performance.
>
> We thank the reviewer for this insightful comment.  To evaluate the robustness of our approach under poor training data, we conducted experiments on Cora, Citeseer, and PubMed by fixing the test set (20%) and varying the size of the training set. The results (shown below) indicate that our model achieves stable performance when trained with at least 50% of the labeled nodes, which demonstrates its effectiveness in supervised settings.
>
> Furthermore, we tested our model on additional datasets (see Table 4) with imbalanced data, such as Tolokers and Question, where class representatives are heavily skewed (e.g., **9192 vs. 2566** and **47461 vs. 1460**, respectively). Despite this imbalance, our model achieves competitive performance compared to GNN and graph transformer baselines. These results suggest that with sufficient random sampling from each class, our contextual embeddings remain effective even in imbalanced scenarios.
>
> | Training % |   Cora   | Citeseer | PubMed |
> |------------|----------|----------|--------|
> | 5%         | 57.38    | 58.56    | 82.12  |
> | 10%        | 70.85    | 66.52    | 84.05  |
> | 20%        | 79.15    | 70.42    | 86.21  |
> | 30%        | 82.29    | 72.67    | 87.04  |
> | 40%        | 83.21    | 76.43    | 87.80  |
> | 50%        | 85.50    | 77.93    | 88.21  |
> | 60%        | 85.95    | 77.63    | 88.26  |
> | 70%        | 87.56    | 79.73    | 88.46  |
> | 75%        | 87.98    | 79.43    | 88.46  |
> ---
>
> ### **W2. Limited Ablation on Landmark Choices and Runtime Breakdown**
> >Limited Ablation on Landmark Choices: Only two landmark types (Euclidean/Jaccard) are explored; impact of number and selection method of landmarks is not fully studied.
>
> We thank the reviewer for this valuable feedback. In our design, the two embeddings serve distinct roles: (1) **Spatial embeddings** are derived from neighborhood label distributions up to a fixed hop size, and do not involve landmarks; (2) **Contextual embeddings** use class landmarks in attribute space, with similarity metrics chosen according to feature type (e.g., positive-overlap/Jaccard for sparse binary features such as Cora and Citeseer, Euclidean for real-valued features such as PubMed). Thus, landmark choices and similarity metrics apply only to contextual embeddings, while spatial embeddings depend on the neighborhood radius.
>
> **Ablation on neighborhood size**. We compared 1-hop vs. 2-hop neighborhoods for spatial embeddings. Accuracy improves modestly with 2-hop, but at increased computational cost (e.g., PubMed 77.59 ± 0.61 / 63s vs. 82.28 ± 0.09 / 232s). Based on this trade-off, we adopt 2-hop neighborhoods in the main experiments.
>
> | **Spatial Only** | Cora (Acc)      | Citeseer (Acc) | PubMed (Acc)   | Cora Time | Citeseer Time | PubMed Time |
> |-----------------|----------------|----------------|----------------|-----------|---------------|-------------|
> | 1-hop           | 83.96 ± 1.35   | 71.28 ± 1.84   | 77.59 ± 0.61   | 1s        | 1s            | 63s         |
> | 2-hop           | 84.61 ± 1.20   | 71.82 ± 1.95   | 82.28 ± 0.09   | 3s        | 3s            | 232s        |
>
> **Ablation on similarity metrics**. For contextual embeddings, we evaluated Jaccard, cosine, and Euclidean metrics. On **binary datasets** (Cora, Citeseer), Jaccard is most effective, while cosine is comparable but more expensive.  As expected, Euclidean distance performs poorly on binary vectors. On **real-valued datasets** (PubMed), Euclidean and cosine both achieve strong results. This confirms that the similarity metric should be aligned with the feature type.
>
> | **Contextual Only** | Cora (Acc)      | Citeseer (Acc) | PubMed (Acc)   | Cora Time | Citeseer Time | PubMed Time |
> |--------------------|----------------|----------------|----------------|-----------|---------------|-------------|
> | Jaccard            | 73.98 ± 2.56   | 67.42 ± 0.85   | -              | 2s        | 7s            | -           |
> | Cosine             | 73.50 ± 2.48   | 67.09 ± 1.80   | 85.62 ± 0.41   | 5s        | 15s           | 17s         |
> | Euclidean          | 62.60 ± 2.65   | 60.89 ± 1.75   | 85.92 ± 0.52   | 2s        | 6s            | 17s         |
>
> Overall, these ablations show that SCNode is **robust to landmark quantity** and that **contextual metrics should be selected according to feature type** to achieve the best balance between accuracy and efficiency. We added these tables and the discussion to Appendix B.4.

---

> ### Author Response · Authors · 2025-08-20
> **Response to Reviewer y9rn-part2**
>
> ### **W3. Heterogeneous Graph Treatment**
> >Scalability to Massive Heterogeneous Graphs: While OGBN-MAG is handled, the approach to collapse heterogeneous relations into paper-only views may lose information and the dimension may be prohibitive in larger knowledge graphs.
>
> We acknowledge the reviewer’s concern. Our current design choice was motivated by two considerations: (1) comparability with prior OGB benchmarks that also focus on the paper subgraph, and (2) highlighting the effectiveness of SCNode embeddings under a simplified and unified homogeneous setting. In fact, our homogeneous graph choice is consistent with prior work [1,2], where heterogeneous relations among authors, venues, and institutions are flattened into a paper-centric view. This setup is widely adopted to ensure comparability across benchmarks and to avoid scalability bottlenecks introduced by relational heterogeneity.
>
> Moreover, our method does not preclude extending SCNODE to heterogeneous graphs. We view the incorporation of relational landmarks or explicit heterogeneous encodings as a promising future direction. Our current focus, however, was to establish strong performance in the widely used homogeneous conversion of OGBN-MAG, consistent with existing baselines.
>
> Reference.
>
> [1] Li et al., Long-range Meta-path Search on Large-scale Heterogeneous Graphs, NeurIPS 24.
>
> [2] Yun et al.  Graph Transformer Networks, NeurIPS 19.
>
> ---
>
> ### **RC4. Benchmark Against Newer Models**
> >Include comparisons to the latest heterophily-focused GNNs (e.g., H2GCN 2024, GraphAny 2024) to ensure novelty.
>
> We thank the reviewers for this valuable suggestion. We have already included results for **H2GCN** as well as other recent baselines such as **TEDGCN**, **FGSAM-SAGE** and **Polynormer** (2024) in our paper (Table 2,3). Regarding the foundation models like **GraphAny** (2024), we note that they adopt a different data splitting strategy for evaluation, which makes a direct and fair comparison infeasible under our current experimental setup.
>
> ---
> ### **RC5. Theoretical Insights**
> >Expand on why the concatenation of spatial/contextual coordinates provably overcomes underreaching, possibly via spectral or expressivity analysis.
>
> Underreaching occurs because a depth-K message-passing GNN is K-local. Its output at node u depends only on the K-hop ego-network and features inside it, so nodes that are K-hop isomorphic are indistinguishable.
> Our inputs break this locality at the feature level through contextual embeddings before any message passing.
>
> **Spectral view.** Spatial coordinates $\alpha_k(u)$ are class-distribution statistics over K-hop neighborhoods and can be written as $\alpha_k = P_k(A)Y$  where $P_k$ is a degree-K polynomial in the normalized adjacency A and Y stacks class indicators of labeled nodes. Any MPGNN of depth K implements a polynomial graph filter in A. In contrast, contextual coordinates $\beta(u)=[d(X_u,\xi_1),\dots,d(X_u,\xi_N)]$ are distances to class landmarks $\{\xi_j\}$ computed in attribute space and are independent of A. Concatenation $[\ \alpha_k(u) || \beta(u)\ ]$ therefore injects a nonlocal, zero-hop channel that no finite-degree polynomial in A can recreate, enlarging the realizable function class.
>
>
> **Expressivity view.** Two nodes that are K-hop isomorphic receive identical inputs in any K-local MPGNN and remain indistinguishable. With contextual embedding $\beta(u)$, such nodes can have different landmark distances if they occupy different positions relative to class prototypes in attribute space, so even a linear classifier on $[\ \alpha_k || \beta\ ]$ can separate them.
>
>
> **Empirical support.** Table 7 shows that contextual-only already resolves heterophilic cases where local neighborhoods are misleading, while spatial-only is strong on homophilic graphs. The combined input wins on every dataset, matching the theory that contextual embedding $\beta$ supplies global class information that a local filter cannot propagate and spatial embedding $\alpha$ exploits reliable local agreement. We added this discussion to appendix (Remark A.5).

---

### Review · Reviewer_7Xoe · 2025-08-08

**Summary Of Contributions:**

This paper proposed a Spatial-Contextual Node Embedding framework (SCNode), which  integrates both spatial and contextual information from local and global scope. It can be easily integrated into any GNN model as plug-and-play components to improve the performance in both homophilic and heterophilic settings. In addition, a new homophily matrices is proposed to understand class interactions and tendencies.

**Audience:**

Yes

**Claims And Evidence:**

No

**Requested Changes:**

1. A case study on low-label rate setting is needed to evaluate the ability of SCNode with tiny number of labels. And it is necessary to investigate whether the effect gain comes from data leakage
2. More strong datasets and baseline methods are needed to show the effectiveness of SCNode.
3. More ablation studies on the plug-and-play component are needed.
4. It it necessary to discuss the connections and difference between other works that analyze the class-wise relations.

**Strengths And Weaknesses:**

Strengths:
1. The spatial and contextual embeddings can capture information from both local and global scopes of a graph.
2. The SCNode vectors can be viewed as plug-and-play components to enhance existing GNN methods.
3. The contextual embedding is interesting while measuring the distance between a node and all class landmarks.

Weaknesses:
1. The SCNode are highly rely on the training set label since the both spatial and contextual node embeddings need label information. It may work badly in real-world low label rate setting. In addition, it seems the calculation of class landmark need all labels rather that training set according to the description, which can leads to data leakage.
2. The datasets and baseline methods are insufficient to evaluate the proposed method. For datasets, the used heterophilic datasets have some drawbacks [1]. For baseline methods, some representative methods [2,3] are not included. In addition, the performance of SCNode on OGB datasets are not satisfactory.
3. As a plug-and-play component, SCNode vectors may bring more computing consumption. How much the performance gain in Figure 4 is attribute to the additional dimensions of node embedding.
4. The claim about “no prior work has proposed a systematic approach to analyze class-wise relations” is incorrect. Here are some works on investigating class-wise relations [4,5,6].

Reference:

[1] A critical look at the evaluation of GNNs under heterophily: Are we really making progress? ICLR 2023.

[2] Revisiting Heterophily For Graph Neural Networks. NeurIPS 2022.

[3] Ordered GNN: Ordering Message Passing to Deal with Heterophily and Over-smoothing. ICLR 2023.

[4] Is Homophily a Necessity for Graph Neural Networks? ICLR 2022.

[5] Heterophily and Graph Neural Networks: Past, Present and Future. IEEE Data Eng. Bull. 2023.

[6] Graph Neural Networks with Heterophily. AAAI 2021.

---

> ### Author Response · Authors · 2025-08-20
> **Response to Reviewer 7Xoe-part 1**
>
> *We sincerely thank the reviewers for their thoughtful reading and constructive feedback, which have greatly helped us improve our paper. A revised version incorporating the suggested changes is available at the link above. Below, we address each concern in detail and highlight the corresponding revisions made to the paper.*
>
> ### **W1. Training Set Label**
>
> > The SCNode are highly rely on the training set label since the both spatial and contextual node embeddings need label information. It may work badly in real-world low label rate setting. In addition, it seems the calculation of class landmark need all labels rather that training set according to the description, which can leads to data leakage.
>
> We thank the reviewer for raising this important concern. We would like to first point out that test data leakage does not exist in our setting. In experiments, the test set was treated as an unknown class and excluded to prevent potential data leakage.
>
> For the data availability question in low-label settings, we conducted additional experiments on Cora, Citeseer, and PubMed, fixing the test set (20%) and varying the training ratio from 5% to 75%. As shown below, SCNode maintains competitive performance, with stable gains once the training set reaches ~50% labels. This demonstrates the effectiveness of our model in a supervised setting.
>
> It is also worth noting that on PubMed, our model performs noticeably well even under low training ratios. This is because, despite only using 5% of nodes for training, the dataset still provides nearly 1,000 labeled nodes across only three classes. This ensures a sufficient number of representatives per class, enabling SCNode to learn effective contextual embeddings. These results have been added to the “Effectiveness in Smaller Training Settings” section.
>
>
> | Training % |   Cora   | Citeseer | PubMed |
> |------------|----------|----------|--------|
> | 5%         | 57.38    | 58.56    | 82.12  |
> | 10%        | 70.85    | 66.52    | 84.05  |
> | 20%        | 79.15    | 70.42    | 86.21  |
> | 30%        | 82.29    | 72.67    | 87.04  |
> | 40%        | 83.21    | 76.43    | 87.80  |
> | 50%        | 85.50    | 77.93    | 88.21  |
> | 60%        | 85.95    | 77.63    | 88.26  |
> | 70%        | 87.56    | 79.73    | 88.46  |
> | 75%        | 87.98    | 79.43    | 88.46  |
>
> ---
> ### **W2. Datasets drawbacks**
>
> > The datasets and baseline methods are insufficient to evaluate the proposed method. For datasets, the used heterophilic datasets have some drawbacks [1]. For baseline methods, some representative methods [2,3] are not included. In addition, the performance of SCNode on OGB datasets are not satisfactory.
>
> We thank the reviewer for raising this important point. To further demonstrate the effectiveness of SCNode, we have expanded our evaluation to include additional heterophilic datasets (see Table 3 in the revised version), such as the *filtered versions* of **Chameleon** and **Squirrel**, as well as **Amazon-Ratings**, **Tolokers**, and **Questions**. SCNode achieves **state-of-the-art performance on two datasets** and delivers competitive results on the others, showing its robustness across diverse graph structures.
>
> On the **OGB benchmarks**, while SCNode does not surpass heavily tuned models with millions of parameters, it remains competitive despite using far fewer parameters. This highlights the efficiency of SCNode as a lightweight yet effective framework. Moreover, we deliberately pair SCNode embeddings with a simple **2-layer MLP classifier** to showcase their standalone strength. As shown in **Table 4**, when SCNode is used as a plug-and-play module with existing GNNs, it consistently delivers **double-digit accuracy improvements**, underscoring its practical value for integration into more advanced GNN or graph transformer architectures.
>
> In addition, we have incorporated **AMC-GCN** and **Ordered-GNN** into our baselines (Table 2) under similar splitting in the revised version to ensure a more comprehensive and representative comparison.

---

> > ### Author Response · Authors · 2025-08-20
> > **Response to Reviewer 7Xoe-part 2**
> >
> > ### **W3. Performance Gain vs Additional Dimension**
> > > As a plug-and-play component, SCNode vectors may bring more computing consumption. How much the performance gain in Figure 4 is  attribute to the additional dimensions of node embedding.
> >
> > We thank the reviewer for this suggestion. To clarify, the performance gains reported in Figure 4 are not solely due to the additional dimensions of SCNode embeddings. In our experiments, we do not add SCNode embeddings, but **replace the original node features with SCNode embeddings**, ensuring that improvements reflect the quality of the spatial and contextual representations rather than simple dimensionality increase.
> >
> > Regarding computational overhead, SCNode embeddings are efficient to compute. For example, on the PubMed dataset (19,717 nodes, 44,338 edges), **spatial embeddings take 232 seconds** and **contextual embeddings take 17 seconds**, which is modest compared to typical training times of GNNs on these datasets. We have added a detailed **ablation study and time breakdown** in the revised ablation section to quantify the contribution of the plug-and-play component to performance and efficiency.
> >
> > ---
> > ### **W4. Class-wise Relations**
> > > The claim about “no prior work has proposed a systematic approach to analyze class-wise relations” is incorrect. Here are some works on investigating class-wise relations [4,5,6].
> >
> > We thank the reviewer for the correction and pointing out these references [4,5,6]. Our original wording was too strong, and we revised it accordingly (Section 2). Prior work does analyze class-wise relations, most commonly through (i) **neighborhood label–mixing statistics** [4,5] and (ii) **class-compatibility matrices** that parameterize inter-class attraction or repulsion [6].
> >
> > Our contribution is complementary but distinct. Rather than relying solely on local neighborhood counts or a global compatibility table, SCNode defines **class-aware coordinates** by triangulating distances to class landmarks. This construction provides each node with **global, long-range class signals** that are not restricted to a fixed hop radius and remain **directly comparable across distant graph regions**. We incorporated these references and clarified this distinction in the revised manuscript (Section 2).

---

### Author Response · Authors · 2025-08-31
**Follow-up on Rebuttal**

Dear Reviewers,

Thank you again for your valuable time and feedback on our paper. We submitted our rebuttal recently and just wanted to kindly check if you have any further questions or comments.

Best regards,

Authors

---

### Comment · Editors_In_Chief · 2025-12-15

On December 9, the authors brought to the attention of the Editors in Chief that the Camera Ready version was still anonymous and had edits marked in blue. On December 13, the authors sent an updated PDF with author identities and coloured text removed, which was uploaded on December 14 by the EiCs. These are claimed to be the only changes.